# Tandemly duplicated CYP82Ds catalyze 14-hydroxylation in triptolide biosynthesis and precursor production in *Saccharomyces cerevisiae*

Yifeng Zhang [1,2,3], Jie Gao[1,2], Lin Ma[2], Lichan Tu [4], Tianyuan Hu [5], Xiaoyi Wu[2], Ping Su[1], Yujun Zhao[1], Yuan Liu[2], Dan Li[6], Jiawei Zhou[7], Yan Yin[2], Yuru Tong[6], Huan Zhao[2], Yun Lu[2], Jiadian Wang[2], Wei Gao [2,3,8] ✉ & Luqi Huang [1] ✉

Triptolide is a valuable multipotent antitumor diterpenoid in *Tripterygium wilfordii*, and its C-14 hydroxyl group is often selected for modification to enhance both the bioavailability and antitumor efficacy. However, the mechanism for 14-hydroxylation formation remains unknown. Here, we discover 133 kb of tandem duplicated CYP82Ds encoding 11 genes on chromosome 12 and characterize CYP82D274 and CYP82D263 as 14-hydroxylases that catalyze the metabolic grid in triptolide biosynthesis. The two CYP82Ds catalyze the aromatization of miltiradiene, which has been repeatedly reported to be a spontaneous process. In vivo assays and evaluations of the kinetic parameters of CYP82Ds indicate the most significant affinity to dehydroabietic acid among multiple intermediates. The precursor 14-hydroxy-dehydroabietic acid is successfully produced by engineered *Saccharomyces cerevisiae*. Our study provides genetic elements for further elucidation of the downstream biosynthetic pathways and heterologous production of triptolide and of the currently intractable biosynthesis of other 14-hydroxyl labdane-type secondary metabolites.

*Tripterygium wilfordii* Hook. F., a medicinal plant also known as Lei Gong Teng, has been used in China for >500 years (Ming dynasty, AD 1476) for the treatment of autoimmune diseases, and modern pharmacological studies have shown its extensive antitumor, anti-inflammatory, and immunosuppressive effects[1,2]. The root is the main medicinal part of *T. wilfordii* and contains up to 415 chemical components, including terpenoids and alkaloids[3]. Among this rich treasure trove of compounds, triptolide (**1**) is undoubtedly an important contributor to pharmaceutical properties[4].

Triptolide, an 18(4 → 3) *abeo*-abietane diterpenoid with a tri-epoxy group and *α,β*-unsaturated lactone moiety (Fig. 1a), has been designed with multiple structure-based artificial derivatives, particularly through carbon-14 (C-14) modifications, which have been implemented to improve its water solubility, and some of the generated compounds

[1]State Key Laboratory of Dao-di Herbs, National Resource Center for Chinese Materia Medica, Chinese Academy of Chinese Medical Sciences, Beijing, China. [2]School of Traditional Chinese Medicine, Capital Medical University, Beijing, China. [3]Beijing Shijitan Hospital, Capital Medical University, Beijing, China. [4]Key Laboratory of Novel Targets and Drug Study for Neural Repair of Zhejiang Province, School of Medicine, Zhejiang University City College, Hangzhou, China. [5]School of Pharmacy, Hangzhou Normal University, Hangzhou, China. [6]School of Pharmaceutical Sciences, Capital Medical University, Beijing, China. [7]College of Biotechnology and Bioengineering, Zhejiang University of Technology, Hangzhou, China. [8]Deceased: Wei Gao. ✉e-mail: weigao@ccmu.edu.cn; huangluqi01@126.com

**Fig. 1 | Bioactive metabolites in *Tripterygium wilfordii* and biosynthesis of triptolide. a** Structures of major bioactive diterpenoids of *T. wilfordii*. **b** Proposed triptolide biosynthetic pathway. A solid arrow represents an identified reaction, and a dotted arrow indicates an unknown pathway. GGPP: Geranylgeranyl diphosphate. **c** Coexpression profiles of CYPs with genes in the triptolide biosynthetic pathway. *TwTPS7(v2)*, *TwTPS27(v2)* and *CYP728B70* are marked in red. The black triangles indicate CYP82D subfamily genes that cluster with identified functional genes. The gene ID of *CYP82D274* is TW12G01155.1. Root periderm (RB), root phloem (RP), root xylem (RX), stem vascular bundle (PS), stem periderm (SB), and leaf (L). Source data are provided as a Source Data file.

have entered clinical trials (ClinicalTrials.gov), leading to a tremendous demand for triptolide. However, the content of **1** in its native plant is only 139 ng·g$^{-1}$ dry weight[5], and the chemical synthesis approach exhibits low synthesis efficiency and multiple steps due to the stereochemical complexity of its structure (e.g., three epoxy groups, unsaturated lactone, and nine chiral centers)[6]. Although the cambial meristematic cells of plant tissue culture technology have increased the yield to 138.1 μg·g$^{-1}$ [7], synthetic biology strategies based on elucidating the biosynthetic pathway appear to be a more promising, sustainable, and alternative method[8,9] and have been successfully applied to achieve heterologous acquisition of potential intermediates and analog triptonide in microorganisms[10–12]. In contrast, natural products such as tripdiolide (**2**), triptolidenol (**3**) and triptriolide (**4**), which have the same 18(4 → 3) *abeo*-abietane skeleton and similar biosynthetic pathways as **1** (Fig. 1a), also have multiple pharmaceutical properties[13]. Elucidating the biosynthetic pathway of **1** will ultimately lead to pharmaceutical, economic and environmental benefits.

The core skeleton cyclization and functional decoration in the biosynthesis of **1** are initiated by two types of diterpene synthases, class II TwTPS7(v2) and class I TwTPS27(v2)[5,14], which cyclize geranylgeranyl diphosphate (GGPP) to produce olefin miltiradiene (**5**) and catalyze double-bond rearrangement on the C-ring for spontaneous conversion to stable aromatized abietatriene (**6**)[8,15], and CYP728B70 then catalyzes carboxylation at C-18 to generate dehydroabietic acid (**7**)[16]. However, the multiple downstream steps from **7** to **1** remain enigmatic and are mainly divided into C-14 hydroxylation, C-18, 19 lactonization, and triepoxidation processes. According to the proposed biosynthetic pathway (Fig. 1b), cytochrome P450s (CYPs) with hydroxylation, epoxidation, and isomerization functions[17] are highly likely to contribute to the biosynthesis of **1**, which has been shown to consist of a minimal set of four biosynthetic components of CYPs for the heterologous production of triptonide[12].

Here, we reveal that CYP82D274 and CYP82D263 in tandem duplicated CYP82Ds catalyze the C-14 hydroxylation grid in the

biosynthesis of **1** through genetic manipulation in native plant cells and functional characterization in a heterologous host. The two CYP82Ds can also transform the aromatization of **5** and promote the production of the rate-limiting **7**. In vivo assays and kinetic parameters indicate that CYP82D274 prefers to catalyze **7** in the pathway and thereby advance the biosynthesis of **1**. The intermediate 14-hydroxy-dehydroabietic acid (**11**) is successfully biosynthesized de novo in *Saccharomyces cerevisiae*. This study provides a systematic introduction to C-14 hydroxylases from evolutionary hypotheses and functional characterization to heterologous biosynthesis and thus paves the way for elucidation of the biosynthetic pathway of **1** and other 14-hydroxyl labdane-/abietane-type secondary metabolites.

## Results

**Screening of candidate CYPs involved in triptolide biosynthesis**
Pathway genes undertake the biosynthesis of secondary metabolites together, and these genes often exhibit similar expression patterns[18–20]; therefore, coexpression analysis is one of the most powerful approaches for the initial screening of functional-associated genes from massive transcriptome data. To identify the CYP gene encoding the enzyme responsible for the biosynthesis of **1**, we analyzed 416 CYPs in the *T. wilfordii* transcriptome (NCBI SRA accession SRP199495) of different plant tissues for expression correlation with previously characterized biosynthetic genes, including *TwTPS7(v2)*, *TwTPS27(v2)* and *CYP728B70*. The expression profiles in the heatmap were grouped into 67 clusters, and 38 genes from 16 CYP families, including CYP71, CYP72, CYP76, and CYP82, exhibited the same pattern of root periderm-specific high expression and clustered with identified functional genes (Fig. 1c). Notably, the most numerous genes, TW12G01155.1 (*CYP82D274*), TW12G01159.1, TW12G01162.1, TW12G01165.1, and TW12G01165.2, belonged to the CYP82D subfamily, whose family is reportedly involved in terpenoid biosynthesis[21,22]. The genes belonging to the CYP82D subfamily were selected as candidate CYPs for further analysis.

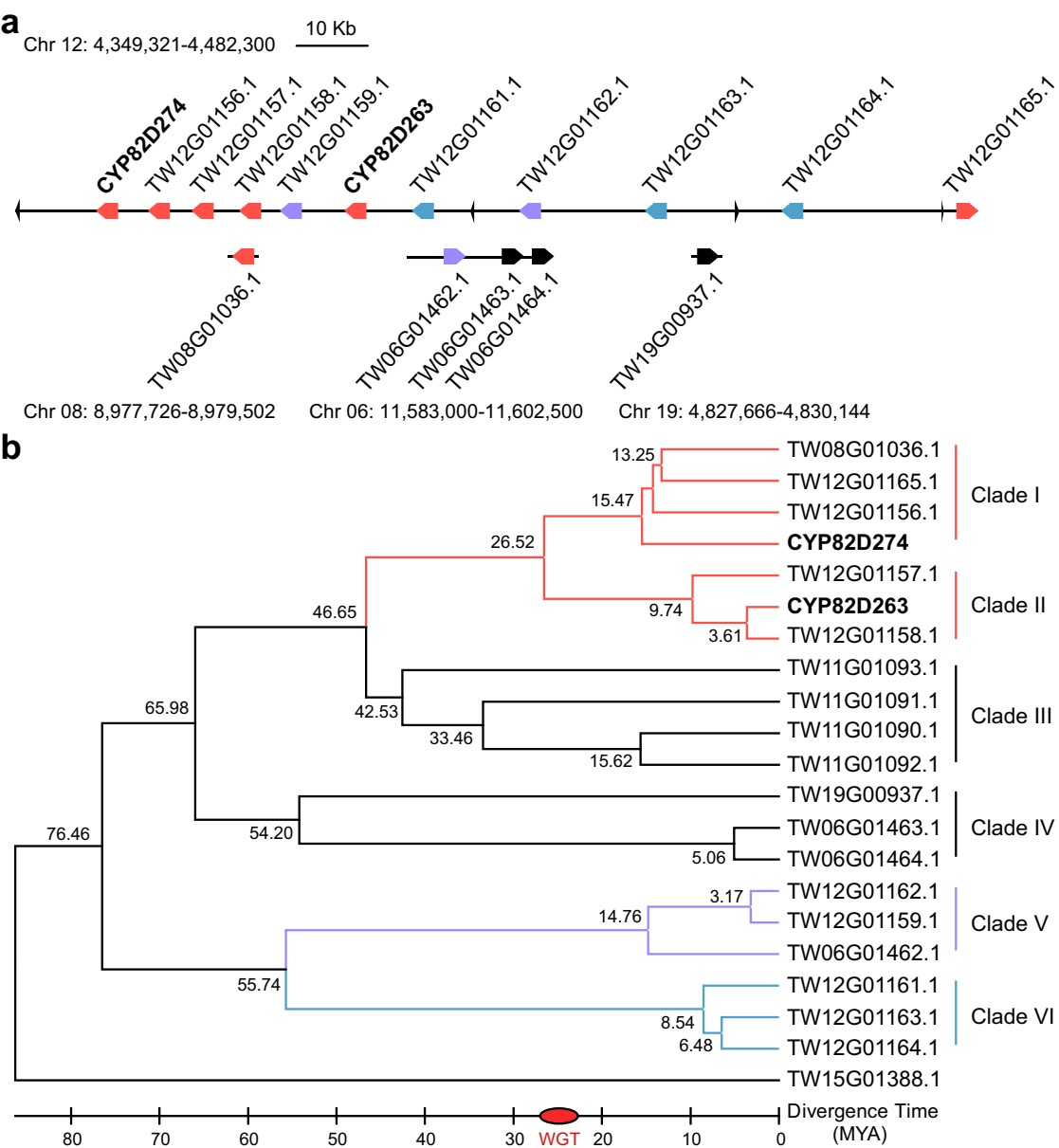

**Fig. 2 | Chromosomal localization and evolutionary analysis of the CYP82D genes. a** Chromosomal localization of the 133-kb tandem duplicated CYP82Ds on Chr12 encoding 11 complete genes. The same color indicates orthologous CYP82D genes and the small triangles represent the incomplete gene residues. **b** Phylogenetic analysis of the TwCYP82D genes. The phylogenetic tree was constructed based on the maximum likelihood method (1000 bootstraps). The numbers represent the predicted divergence time and WGT indicates the whole-genome triplication event of *T. wilfordii*.

## Tandem duplication of CYP82Ds and analysis of their transcriptional expression

A chromosomal localization analysis of all CYP82D genes revealed that they were localized at Chr06, Chr08, Chr11, Chr12, Chr15, and Chr19. Further local BLAST analysis of the whole genome using the nucleic acid and amino acid sequences of CYP82D genes revealed a tandem duplication gene cluster at Chr12 containing 11 complete CYP82D genes, which had a length of 133 kb with 4 additional incomplete CYP82D gene residues (Fig. 2a). This tandem duplication of CYP82Ds occurred in dynamic chromosomal regions enriched in transposable elements (TEs) and was presumably generated by the duplication and rearrangement of long terminal repeat (LTR) elements of retrotransposons within chromosomes. A phylogenetic analysis of orthologous CYP82D genes (Fig. 2b) indicated that all TwCYP82Ds clustered into six clades, and the duplication event and neofunctionalization of Clade I and Clade II appeared to have occurred at 26.52 MYA, which

was comparable to the timing of the *T. wilfordii* whole-genome triplication (WGT) event[16]. The synonymous substitution rate (*Ks*) values of orthologous CYP82D genes indicated that the specific expansion of the CYP82D genes occurred via tandem duplication (Supplementary Data 1).

Heatmap analysis of the genes in Clade I and Clade II with known functional genes *TwTPS7(v2)*, *TwTPS27(v2)*, and *CYP728B70* as well as the content of **1** was achieved by combining the transcriptomic data and metabolite analysis (Supplementary Fig. 1). **1** exhibited a distribution pattern of root periderm and leaf tissue-specific expression. *CYP82D274* and TW12G01165.1 in Clade I showed the same high expression pattern in root tissues as the previously characterized genes, whereas *CYP82D263* in Clade II showed high expression in the root periderm, stem periderm, and leaves. TW12G01158.1 was highly expressed in the stem periderm and leaves, particularly the stem periderm. In addition, TW12G01156.1 and TW12G01157.1 were not

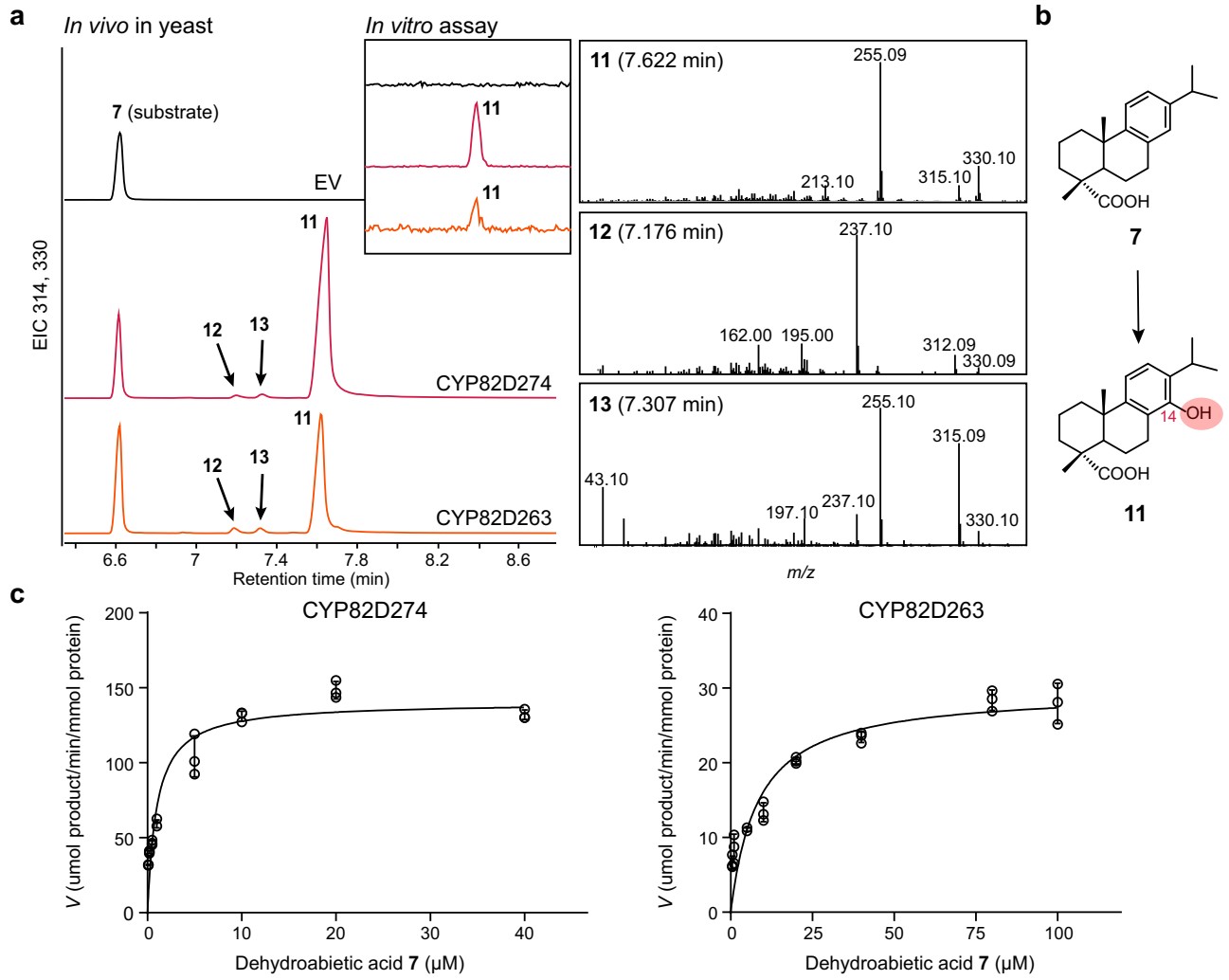

**Fig. 3 | Functional characterization of CYP82D274 and CYP82D263. a** GC–MS analysis of methylated products of CYP82D274 and CYP82D263 catalyzing dehydroabietic acid (**7**) in vivo or in vitro. Empty vector (EV) denotes yeast transformed with an empty vector without CYP. **b** Catalytic process in **a**. **c** Kinetic profiles of CYP82D274 and CYP82D263 catalyzing **7** in vitro. The quantification of **11** was based on the standard curve $y = 0.7336x - 0.014$ ($R^2 = 0.9994$) obtained by LC-TQ-MS/MS.

The concentration of CYPs was estimated by measuring the reduced CO-difference spectrum. Kinetic parameters were calculated by nonlinear regression analysis using the Michaelis–Menten model. Data are presented as mean values ± standard deviation SD from three biological independent replicates, and the black circles represent the individual data points. Source data are provided as a Source Data file.

detected and were supposedly selectively silenced by gene redundancy during evolution. In particular, TW08G01036.1 (*CYP82D213*) exhibited root peridermal-specific expression, and CYP82D213 is reportedly involved in the final step of the analog triptonide biosynthetic pathway. The transcriptional expression of the other five complete CYP82D genes of this tandem duplication gene cluster was also analyzed. In Clade V, TW12G01162.1 was highly expressed in the root and stem periderm, whereas TW12G01159.1 had a base deletion at position 44 in the coding region, causing termination of protein translation at position 26. TW12G01164.1 in Clade VI was highly expressed in the stem periderm, whereas the expression of TW12G01161.1 and TW12G01163.1 was too low to be detected.

## CYP82D274 and CYP82D263 catalyze C-14 hydroxylation of dehydroabietic acid

Among the CYP82D genes discovered from the transcriptomes, 11 were cloned (Supplementary Table 1). To investigate the biochemical activity of the CYPs, these 11 CYP82Ds were integrated into the plasmid pESC-LEU expressing both CYP and cytochrome P450 oxidoreductase 3 from *T. wilfordii* (TwPOR3) and then transformed into the yeast strain BY4741. Substrate feeding has been proven to be effective in yeast

fermentation[16,23]; thus, the intermediate **7** was fed into the cultures, and the fermentation products were extracted. Among 11 CYP82Ds, CYP82D274 and CYP82D263 accepted **7** as a substrate with the same three products at *m/z* 330 (Fig. 3a and Supplementary Fig. 2). The major product **11** was enriched and identified as 14-hydroxy-dehydroabietic acid (**11**), which was independently confirmed by ${}^{1}$H, ${}^{13}$C, HSQC, ${}^{1}$H-${}^{1}$H COSY, and NOESY NMR analyses (powder purity 98.23%) (Supplementary Figs. 3-7 and Supplementary Note 1). The additional minor product **13** was identified as 15-hydroxy-dehydroabietic acid upon comparison to the authentic standard (Supplementary Fig. 8). And further optimization of the GC–MS detection by increasing the MS resolution revealed that CYP82D274 and CYP82D263 catalyzed the production of another byproduct, 12-hydroxy-dehydroabietic acid (**18**)[24], which is isomeric with **11** and has identical MS fragments (Supplementary Fig. 9). Altogether, CYP82D274 and CYP82D263 can catalyze **7** to produce **11** by inducing C-14 hydroxylation (Fig. 3b), albeit with promiscuity (C-12, C-15, etc.).

As resulted by in vitro enzyme assays, both CYP82D274 and CYP82D263 could exhibit their major 14-hydroxylase functions and generated **11** in the presence of TwPOR3, NADPH, substrate **7**, and cofactors (Fig. 3a). The $K_m$ values of CYP82D274 and CYP82D263 were

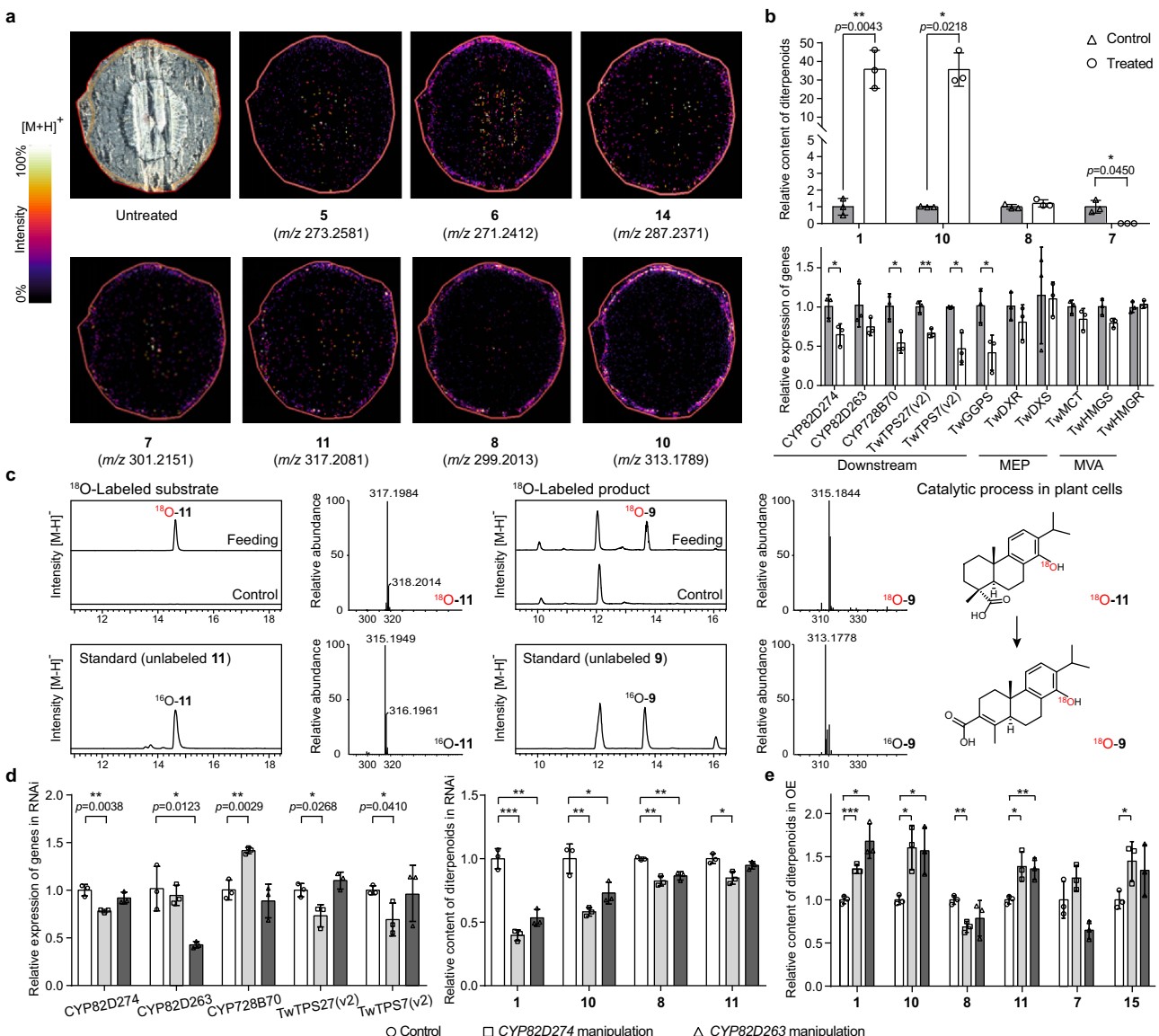

**Fig. 4 | Intermediates and genes involved in triptolide biosynthesis. a** MALDI-MSI analysis of the distribution of triptolide intermediates in *T. wilfordii* root tissue. DHB was used as a matrix for metabolite imaging in the positive ion mode. The colors represent the intensity percentage, and each image is independent. **b** Relative content of diterpenoids and relative quantification of gene expression in 14-hydroxy-dehydroabietic acid (**11**)-fed cell lines. Cells that were not fed **11** served as a negative control. *P*-values of genes with significant differences in expression between groups were 0.0375, 0.0178, 0.0031, 0.0448, and 0.0294 in that order. **c** Metabolite analysis of 18O-labeled **11**-treated cell samples. 18O-**11** was fed to *T. wilfordii* suspension cells that inhibited the biosynthesis of precursors, and wild-type (WT) and unlabeled **11**-treated cells were used as negative controls. 18O-**11** was converted into 18O-triptinin B (**9**) in plant cells, and the mass spectra of labeled versus unlabeled metabolites are provided. **d** Relative content of diterpenoids and relative gene expression in *CYP82D274* and *CYP82D263* RNAi cell lines. **e** Relative content of diterpenoids in *CYP82D274*- and *CYP82D263*-overexpressing cell lines. The cell lines bombarded with corresponding empty vectors served as controls in **d** and **e**, and significant differences (*P*-values) in metabolites between the groups are shown in Supplementary Data 2 and 3. **b**, **d**, **e** The relative quantification of each metabolite was calculated by dividing each sample by the average content of the control group. The relative expression of genes was determined by the $2^{-\triangle\triangle Ct}$ method. *EFLα* was designated as the housekeeping gene, and the corresponding control group was assigned as the reference sample. Data are presented as mean values ± SD (*n* = 3 biologically independent replicates). \*\*\**P* < 0.001, \*\**P* < 0.01 and \**P* < 0.05 determined by two-sided Student's *t* test. Source data are provided as a Source Data file.

0.99 ± 0.17 μM and 8.42 ± 1.89 μM, respectively, at the optimal reaction time (Fig. 3c and Supplementary Fig. 10a). These steady-state kinetic parameters indicated that CYP82D274 was significantly superior to CYP82D263 in terms of substrate affinity and catalytic efficacy.

### *CYP82D274* and *CYP82D263* are involved in triptolide biosynthesis

To further investigate the definitive role of *CYP82D274* and *CYP82D263* in **1** biosynthesis, RNA interference (RNAi) was utilized to knock down the expression of these two genes in *T. wilfordii* suspension cells. All

TwCYP82D nucleic acid sequences in the genome showed a high identity of 69.61%, mainly between CYP82D274 and CYP82D263, with 81.41% concordance. We specifically selected a 459-bp (nucleotides 459-917) fragment of *CYP82D274* and a 498-bp (nucleotides 373-870) fragment of *CYP82D263* to construct the binary vector pK7GWIWG2D (II) and transformed them into suspension cells by plasmid bombardment[25]. The electrophoretic bands of the vector-specific fragment indicated successful transformation (Supplementary Fig. 11a), and detection of the expression of *CYP82D274* and *CYP82D263* showed that they were targeted for disruption as expected (Fig. 4d).

Metabolite profiling revealed that the inhibition rate of **1** was 60.37% and a considerable reduction in the accumulation of triptophenolide (**10**), triptobenzene D (**8**) and the direct product **11** in the *CYP82D274*-RNAi cell lines (Fig. 4d and Supplementary Data 2). In particular, the expression of *CYP728B70*, *TwTPS27(v2)*, and *TwTPS7(v2)* exhibited opposing changes, which were hypothesized to be intergenic feedback regulation. In the *CYP82D263*-RNAi cell lines, three metabolites, **1**, **10** and **8**, were significantly decreased. Furthermore, based on the inhibition rates of individual metabolites, we observed stronger inhibition of metabolites in *CYP82D274*-RNAi and *CYP82D263*-RNAi as the pathway progressed downstream, suggesting a cascade accumulation of metabolites.

*CYP82D274* and *CYP82D263* were also specifically overexpressed in plant cells, as evidenced by electrophoretic bands and gene expression (Supplementary Fig. 11). Targeted metabolite analysis indicated that the overexpression of *CYP82D274* and *CYP82D263* resulted in a significant increase in the accumulation of **1** and **10** as well as the direct product **11** (Fig. 4e and Supplementary Data 3). The results of these in vivo assays suggested that CYP82D274 and CYP82D263 are involved in **1** biosynthesis as 14-hydroxylases.

### 14-Hydroxy-dehydroabietic acid is a precursor of triptolide

To provide direct evidence showing whether **11** is implicated in **1** biosynthesis, mass spectrometry imaging and metabolite analysis were employed. Previous studies found that **1** exhibits a build-up property specific to root periderm tissues (Supplementary Fig. 1). The distribution of metabolites in fresh roots was investigated in a targeted manner using MALDI-TOF-MS, and the acquainted intermediates **6** and **10** were found to accumulate within the periderm (Fig. 4a). In addition, the putative intermediates **7** and **8**[26], including the compound **11** discovered in this study, with carboxyl groups also displayed a high distribution in the periderm region, elucidating their spatial correlation. However, **1** was not available for MALDI-MS imaging due to the extremely low content of the compound, the choice of medium, the ionic pattern, and the competition of surrounding ions for laser ionization possibilities.

In combination with the abovementioned tissue distribution pattern of the intermediates, we fed **11** to *T. wilfordii* suspension cell cultures and found that the contents of **1** and **10** were elevated 35.73 and 35.60 times, respectively, compared with the levels in the negative control (Fig. 4b and Supplementary Data 4). Moreover, the catalytic

substrate **7** of CYP82D274 and CYP82D263 was markedly reduced. The results further showed that all downstream pathway genes known to be involved in **1** biosynthesis (*CYP82D274*, *CYP82D263*, *CYP728B70*, *TwTPS27(v2)*, *TwTPS7(v2)*, and *TwGGPS*) showed reductions in expression to 0.42-0.76-fold in different proportions but did not affect genes involved in the upstream MVA and MEP pathways (Fig. 4b). To ensure that the abovementioned changes in metabolites were caused by **11**, stable isotope labeling was employed for more in-depth analysis. The use of the MEP inhibitor fosmidomycin and the use of the gene gun to suppress *TwTPS7(v2)* and *TwTPS27(v2)* inhibited the biosynthesis of precursors, whereas the elicitor subsequently increased the expression of downstream genes, as revealed when [18]O-**11** was fed to the cells. Although the native metabolic pathway was not completely inhibited and endogenous **1** and **10** interfered with the determination of isotopically labeled products, we still found the peak of [18]O-labeled triptinin B (**9**) (Fig. 4c), which could be further lactonized by CYP71BE to generate the expected intermediate **10**[12]. The above-described results indicated that **11**, which has a carboxyl group, serves as a precursor in the late stage of **1** biosynthesis.

### CYP82D274 and CYP82D263 catalyze the hydroxylation and aromatization of multiple substrates

When *CYP82D274* and *CYP82D263* were overexpressed, a 0.69-fold decrease in **8** and a 1.45-fold increase in ferruginol (**15**) were also observed, which appeared to indicate other catalytic processes played by these two genes (Fig. 4e and Supplementary Data 3). The fermentation of yeast strains harboring CYP82D274 or CYP82D263 and subsequent incubation with the intermediate **8** revealed that CYP82D274 generated three hydroxylated products with a charge of $m/z + 16$ (Fig. 5a). The retention time and mass spectral fragmentation for the major product matched the authentic standard for the derivatization of **9** (Supplementary Fig. 12), indicating that CYP82D274 played a C-14 hydroxylation role in catalyzing the formation of **9** from **8**. In addition, the incubation of substrate **5** further gave rise to two different ratios of C-14 and C-12 hydroxylated products (Fig. 5b and Supplementary Fig. 13). For CYP82D274, the major product **14** was identified to be 14-hydroxy-abietatriene with small amounts of **15** (12-hydroxy-abietatriene or ferruginol, Supplementary Fig. 14). For CYP82D263, **5** was converted into the hydroxylated major product **15** accompanied by trace amounts of **14**.

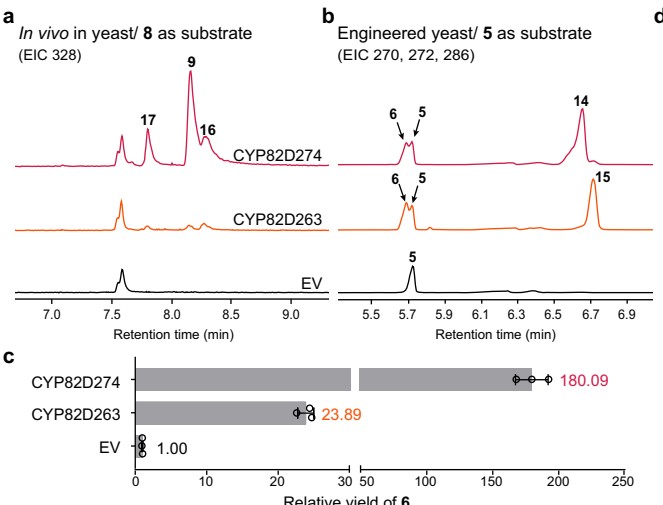

**Fig. 5 | Metabolic grid of C-14 hydroxylation and aromatization. a** GC–MS analysis of methylated products with triptobenzene D (**8**) as the substrate. **b** GC–MS analysis of the catalytic products of miltiradiene (**5**) in engineered yeast. The mass spectrum results are shown in Supplementary Figs. 12 and 13. **c** Relative yield of abietatriene (**6**) in spontaneous, CYP82D274 and CYP82D263 cultures. Data are

presented as mean values ± SD (*n* = 3 biologically independent replicates). **d** Catalytic process of the CYP82D274 and CYP82D263 metabolic grid for triptolide (**1**) biosynthesis. A solid arrow represents an identified reaction, and a dotted arrow indicates an unknown reaction. Source data are provided as a Source Data file.

Double bond rearrangement in the C-ring of olefin **5** to yield stable aromatized **6**[27] has been repeatedly reported to be a spontaneous reaction process[14,28–30]. In our study, we transformed plasmids into engineered yeast BY-PS2[16], which could self-produce **5** as a substrate. The presence of a double peak for **5** and **6** in CYP82D274 and CYP82D263 compared with the single peak for **5** in the EV (Fig. 5b) indicated that CYP82D274 and CYP82D263 were capable of catalyzing the aromatization of **5** to generate **6**. Nevertheless, we demonstrated the existence of spontaneous reactions with small amounts of **6** (Fig. 5c), and if the yield of **6** in the EV was regarded to be 1.00, the relative yield of **6** in CYP82D274 was 180.09 and that in CYP82D263 was 23.89. This finding suggested that CYP82D274 and CYP82D263 can specifically and effectively catalyze the aromatization of **5**, whereas other genes in the same CYP82D subfamily cannot (Supplementary Fig. 15).

Based on these results, we conclude that CYP82D274 and CYP82D263 catalyze important hydroxylation and aromatization processes and contribute to the biosynthesis of **1** through multiple pathways (Fig. 5d).

### CYP82D274 creates a metabolic grid in triptolide biosynthesis

In the previous section, we revealed that the wide substrate range of CYP82D274 enabled the simultaneous catalysis of different substrates, and CYP728B70 catalyzes the reaction of **5** to generate **7**[16]. By feeding strains harboring CYP728B70 with **14**, the fermentation products were detected to contain both abietatriene-14,18-diol (a CYP728B70 catalyzed intermediate, **19**) and **11** (Supplementary Fig. 16)[31], suggesting that the biosynthetic pathway is not a linear and specific pathway but rather that the products are generated via multiple pathways together through a metabolic grid, a phenomenon that has also been found in forskolin and tanshinones[32,33].

In the face of a metabolic grid with multiple substrates, the substrate affinity and catalytic efficiency of the enzymes are quite different and can be reflected by kinetic parameters[34]. We found that the $K_m$ values of CYP82D274 and CYP82D263 for catalyzing **7** equaled $0.99 \pm 0.17\,\mu M$ and $8.42 \pm 1.89\,\mu M$ (Fig. 3c), indicating their higher sensitivity to **7**. We also carried out microsomal experiments with CYP82D274 and CYP82D263 in the presence of NADPH, a redox partner, cofactors, and the substrates **5** and **8**. The results showed that only CYP82D274 could catalyze **5** to produce small amounts of **14** with a $K_m$ of $47.30 \pm 12.98\,\mu M$ (Supplementary Fig. 10b), but the target product was not detected under the conditions of catalytic **8** or CYP82D263-catalyzed **5**. In addition, the $V_{max}$ values of CYP82D274 for catalyzing **7** and **5** were $140.30 \pm 5.10$ umol of product·min$^{-1}$·(mmol of protein)$^{-1}$ and $44.08 \pm 6.13$ umol of product·min$^{-1}$·(mmol of protein)$^{-1}$, these kinetic parameters indicated that CYP82D274 has a higher preference for substrate **7**.

### De novo biosynthesis of 14-hydroxy-dehydroabietic acid in yeast

To further address the unknown biosynthetic pathway of **1**, self-produced engineered yeast was first constructed to provide sufficient amounts of the precursor **11** for gene mining. CYP82D274 and CYP82D263 were coexpressed with CYP728B70, suggesting that yeast is capable of expressing multiple CYPs simultaneously to produce **11** (Fig. 6a and Supplementary Fig. 17). To achieve a high yield of **11**, three functional modules were designed and investigated for optimization. Multispecies-derived diterpene synthases confirmed that tSmKSL1-CfTPS1 (the fusion of truncated *ent*-kaurene synthase-like 1 from *Salvia miltiorrhiza* and terpene synthase from *Coleus forskohlii*) was the most efficient **5** biosynthase[10]. The GGPP high-yielding strain BY-HZ16[10], tSmKSL1-CfTPS1 (module I) and the more active CYP82D274 (module III) were selected for the production of **11**. In this section, we utilized synthetic biology strategies for the optimization of genetic elements and chassis strains.

The optimization of protein–protein interactions between CYP and its redox partner POR to enhance the electron transfer efficiency can alleviate the poor coupling of CYP systems and increase terpenoid production[35]. Because module II (generation of **7**) acted as a rate-limiting step hindering the final product yield, this module was first optimized for the genetic elements. Comparison of the yields from multiple species sources revealed that native POR generally exhibits the highest efficiency of electron transfer[36,37]. Twelve enzyme combinations (Supplementary Table 2) with truncated or nontruncated CYP728B70 and representative TwPORs[38] (without TwPOR2, a high identity of 98.73% with TwPOR1) were transformed into the BY-ZY1 strain. LC–MS/MS quantification of the yield of **11** showed that TwPOR3 and TwPOR4 with intact CYP728B70 had the highest yields among the combinations, presumably due to differences in the electron transfer capacity associated with the phylogenetic clade of POR (Supplementary Fig. 18)[39]. None of the six combinations with truncated TwPORs detected the final product, suggesting that the intact transmembrane domain is essential for electron transfer. In addition, truncated CYP728B70 showed reduced yields, indicating that the complete transit peptides are important for peptide translocation and folding. TwPOR3, which is more biostable, was selected as the electron shuttle with a yield of **11** equal to 2.23 µg·L$^{-1}$ (Fig. 6b). Further replacement of CYP728B70 with CYP720B4 from *Picea sitchensis* resulted in a 17.10-fold increase in the yield to 49.43 µg·L$^{-1}$ (Fig. 6b).

Subsequently, optimization of the chassis strain was performed. Chromosome integration and diploidization are promising strategies to improve heterologous gene stability and expression levels and thus enhancing the fermentation ability[40,41]. We introduced the optimal genes (i.e., *tSmKSL1-CfTPS1*, *CYP82D274* or mutants, *CYP720B4*, and *TwPOR3*) into the diploid strain BY-ZY2D (Supplementary Table 3) and haploid strain BY-HZ16. Diploid strains have higher cell growth rates, cell yields, and tolerances to various stresses than haploid strains[42]. In this study, we quantitatively inoculated different haploid and diploid strains and confirmed that the diploid strains exhibited ~2–3 times more biomass than the haploids under identical fermentation conditions (Fig. 6c). The diploid strain BY-ZY16D, in which *tSmKSL1-CfTPS1* and *CYP82D247[L234M]* were integrated into the yeast chromosome and which carried an inducible plasmid pESC-LEU::(*CYP720B4 + TwPOR3*), had the highest yield among the compared strains, and the yield of **11** equaled 343.87 µg·L$^{-1}$, which was 1.34 times ($P < 0.05$) higher than that of the WT BY-ZY15D strain and 6.78 times ($P < 0.01$) higher than that of the haploid strain BY-ZY12H of the same genotype (Fig. 6c).

## Discussion

Triptolide (**1**) is largely insoluble in aqueous solvents; however, chemical structure-bioactivity correlation analyses indicate that the characteristic hydrogen bond at the C-14 position is the key functional group for its antitumor effect by selective alkylation of thiol groups of enzyme-mediated tumor growth[13,43,44]. Further modification of the C-14 position, which was the earliest and most diverse target, is effective not only for increasing its water solubility[45] but also for enhancing its antitumor activity and lower toxicity[44]; as a result, various compounds, e.g., minnelide (ClinicalTrials.gov: NCT03129139) and 14-succinyl triptolide sodium salt (PG490-88), have entered clinical trials. The biosynthetic pathway of **1** in native plants has advanced to the first step in skeleton modification after several years[5,14,16]. In this study, we revealed the mechanism for 14-hydroxylation formation and demonstrated the involvement of carboxyl groups in the biosynthetic pathway of **1**. CYP82D274 and CYP82D263 act as 14-hydroxylases to catalyze the metabolic grid in **1** biosynthesis and exhibit better affinity and catalytic efficiency for multiple intermediates toward **7** (Fig. 5d). Previous pharmacological studies on the chemical synthesis of **11**, the main product of CYP82D274 catalyzing **7**, have shown its small level of cytotoxic activity, particularly in human acute T-cell leukemia[31]. Using

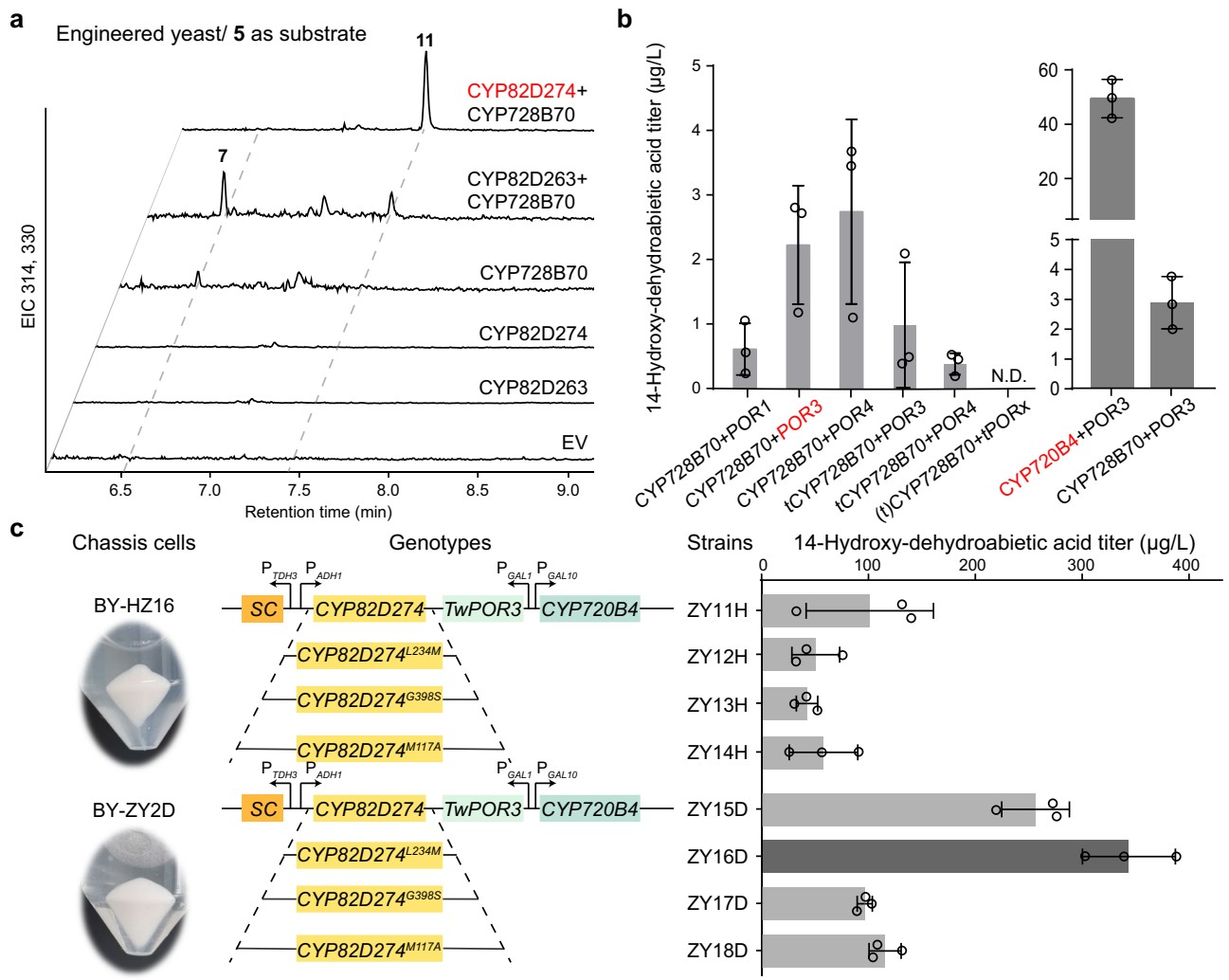

**Fig. 6 | De novo biosynthesis of 14-hydroxy-dehydroabietic acid (11) in yeast.**
**a** GC–MS analysis of methylated products in CYP82D274 and CYP82D263 coexpressed with CYP728B70. **b** Screening for optimal combinations of CYP and TwPOR. Optimal genes are marked in red. N.D. indicates not detected. **c** Yield of **11** in the engineered strains. The images show the biomass of haploid and diploid strains under the same culture conditions. The genotype schematic is shown in the figure, and detailed information is provided in Supplementary Table 3. Data are presented as mean values ± SD ($n = 3$ biologically independent replicates). Source data are provided as a Source Data file.

synthetic biology strategies, the intermediate **11** was successfully produced in yeast with a shake flask yield of 343.87 µg·L$^{-1}$ to not only obtain pharmacologically active products in an environmentally friendly manner but also facilitate the mining of enigmatic pathway genes.

In particular, CYP82D274 and CYP82D263 can catalyze the aromatization of **5** to yield **6** (Fig. 5), a process that was previously reported as a spontaneous reaction[8,15,28], which results in the widespread presence of tanshinones, carnosic acid, carnosol, and other labdane- or abietane-type diterpenoids. We have demonstrated that functionalized CYP82D274 and CYP82D263 could increase the yield of the rate-limiting enzyme and thus the heterologous production of final products (Fig. 6a). It is reasonable to believe that the introduction of CYP82D274 and CYP82D263 in other **5**-derived metabolic pathways could effectively improve access to the target products. Furthermore, the extensive C-14 hydroxylation function of CYP82D274 could effectively provide the necessary alternative genetic element and synthetic precursors for these pharmacologically important natural products and derivatives, for instance, gerardianin A from *Isodon lophanthoides*[46].

CYP82D274 and CYP82D263 identified in this study are located on a tandem duplication gene cluster on Chr12 containing 11

CYP82D subfamily genes (Fig. 2a), presumably arising from LTRs of the retrotransposon (Class I element) generated by intrachromosomal replication rearrangements. According to the evolutionary divergence, the following hypothesis is proposed. In Clade I, the first duplication of neofunctional CYP82D produced CYP82D274, and gene duplication and transfer of TW08G01036.1 (CYP82D213) via transposon from Chr12 to Chr08 then occurred; in addition, the duplication of CYP82D274 produced TW12G01156.1, which then duplicated to generate TW12G01165.1 rearrangement at the other end of the tandem array. CYP82D213 reportedly undertakes the triepoxidation function in the last step of triptonide[12], indicating that it underwent neofunctionalization after separation from CYP82D274. In Clade II, the duplication order follows TW12G01157.1, TW12G01158.1 and CYP82D263 (Supplementary Fig. 19).

Genes in the CYP82D subfamily are commonly found in flavonoid biosynthesis. SbCYP82D1.1 in *Scutellaria baicalensis* functions as a flavone 6-hydroxylase (F6H) in catalyzing chrysin, whereas SbCYP82D2 acts as a flavone 8-hydroxylase (F8H)[47]. A comparison of eight orthologous gene pairs of CYP82Ds in the genomes of *S. barbata* and S. *baicalensis* revealed the presence of the F6H tandem gene clusters *SbaiCYP82D1-SbaiCYP82D7-*

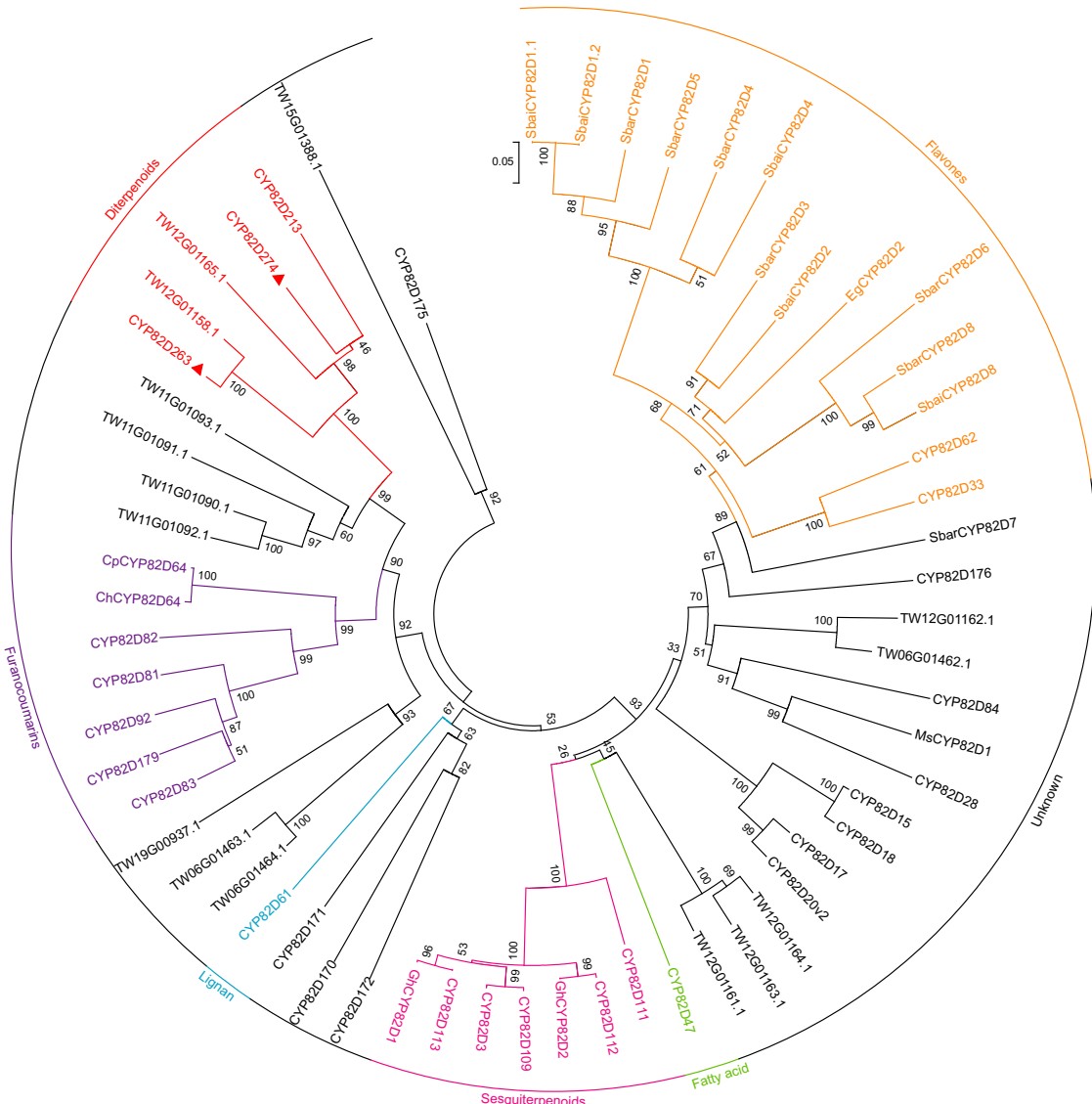

**Fig. 7 | Phylogenetic analysis of CYP82Ds.** A total of 61 CYP82Ds from flavonoids, furanocoumarins, lignan, fatty acid, sesquiterpenoids, diterpenoids, etc., were included and indicated by different colors. Phylogenetic analysis was performed using MEGA 6.0 software with maximum likelihood method (1000 bootstraps). The two functional CYP82Ds in this study are marked with red triangles. The GenBank accession numbers are provided in Supplementary Data 6.

*SbaiCYP82D8* and *SbarCYP82D1-SbarCYP82D6-SbarCYP82D8* on Chr06 in both species, with chromosomal localization distances of <30 kb. In particular, CYP82D8 showed interspecies gene synteny, indicating conserved and species-specific gene evolution[48]. In addition to flavonoids, CYP82Ds have been reported to be involved in the biosynthesis of furanocoumarin isopimpinellin[49], lignan (-)−4'-desmethylepipodophyllotoxin[50], and sesquiterpenoid gossypol[21]. Both CYP82D213 and the two CYP82D genes reported in the present study are involved in diterpenoids biosynthesis, and the construction of a phylogenetic tree with different functions (Fig. 7) revealed a clear phylogenetic differentiation of CYP82Ds involved in different types of compounds, which can also be used to guide future directions of research on CYP82Ds of unknown function.

Recently, a minimal set of four genes from the CYP71BE and CYP82D subfamilies required for the formation of triptonide from **5** was reported[12], and these undoubtedly constitute an alternative and simple way to bypass the carboxylation step and obtain triptonide in heterologous hosts. However, whether it is an absolute biosynthetic pathway or a branched metabolic pathway in its native plants remains unclear. Here, we confirmed the involvement of carboxyl groups in the biosynthetic pathway of **1** via plant cell experiments. From the current perspective, C-18 carboxyl generation is the rate-limiting step responsible for the extremely low levels of **1** in *T. wilfordii*, and the *Tripterygium* genus contains many active natural products with carboxyl or carbonyl groups at C-18 or C-19[3,26]. Improving protein activity is essential for improving the yield of carboxyl-related metabolites. The mechanism of carboxyl shift also needs to be explored for the biosynthesis of **1**.

In conclusion, our study revealed the key enzymes for the C-14 hydroxylation metabolic grid of **1**, characterized the hydroxylation and aromatization functions of tandemly duplicated CYP82D274 and CYP82D263, and successfully achieved the de novo biosynthesis of precursor **11** in yeast. The elucidation of the formation mechanism of C-14 hydroxylation not only provides important genetic elements for the synthetic biology production of **1** but is also useful for revealing the biosynthetic pathways of other labdane-type compounds with C-14 hydroxylation.

## Methods

### Plant materials and chemicals

*T. wilfordii* suspension cells were cultured in Murashige & Skoog (MS) with vitamins (Caisson Lab, USA) medium containing 30 g·L$^{-1}$ sucrose with 0.5 mg·L$^{-1}$ indole-3-butyric acid (IBA), 0.5 mg·L$^{-1}$ 2,4-dichlor-ophenoxyacetic acid (2,4-D) and 0.1 mg·L$^{-1}$ kinetin (KT), and the final pH was adjusted to 5.8. Suspension cells were grown at 25 °C in dark-ness with shaking at 120 rpm[51]. A total of seven four-year-old *T. wilfordii* plants (No. 164–170) were collected from Taoyuan Town, Datian County, Sanming City, Fujian, China. The geographic coordinates were E117°31′44″, N25°49′29″, and the altitude was 908 m. The chemical standards **7**, **10**, and **13** were purchased from Shanghai Yuanye Bio-Technology Co., Ltd. (Shanghai, China), and **9** and **15** were procured from BioBioPha Co., Ltd. (Yunnan, China). Compound **8** was kindly provided by Prof. Fei Li from Kunming Institute of Botany, Chinese Academy of Sciences, and **5**, **6**, **11**, and **14** were isolated from enriched fermentation products.

### Strains, vectors and media

The initial yeast strain used in this study was BY-HZ16 (*MATα*; *trp1Δ0*; *leu2Δ0*; *lys2Δ0*; *ura3Δ0*; *rox1Δ*; *yjl064wΔ*; *ypl062wΔ*; *erg9::Δ−218−−17S*; *trp1::HIS3*-P$_{PGK1}$-BTS1/ERG20-T$_{ADH1}$-P$_{TDH3}$-SaGGPS-T$_{TPI1}$-P$_{TEF1}$-tHMG1-T$_{CYC1}$)[10]. The engineered yeast strains are listed in Supplementary Table 3. *Escherichia coli* Trans1-T1 (TransGen Biotech, China) was used for cloning and plasmid construction. The shuttle vectors were pESC-LEU, pESC-TRP (Agilent Technologies, USA), and pYES2 (Invitrogen, USA). Synthetic dropout (SD) medium (FunGenome, China) was selected according to the auxotroph and carried plasmids. Yeast extract peptone dextrose (YPD) and YPL media were composed of 1% yeast extract, 2% peptone (OXOID, UK), and 2% D-(+)-glucose (Glc, Sigma–Aldrich, USA) for YPD or 2% D-(+)-galactose (Gal, Inalco S.p. A, Italy) for YPL.

### Chromosome location and evolution analysis of CYP82Ds

To obtain more chromosome information on CYP82Ds, BioEdit software (version 7.0.9.0) was used to search the open reading frame (ORF) of CYP82Ds from the whole genome through the local BLAST function. A sequence with a 'blastn' identity greater than or equal to 99% was regarded as one gene, and genes with similarity greater than 50% were recorded. Each CYP82D was ultimately identified by chromosome localization. A phylogenetic analysis was performed using MEGA 6.0 software to align the CYP82D sequences using ClustalW function, and the maximum likelihood method (1000 bootstraps) was used to build a phylogenetic tree. A total of 3n different sites were removed, and the online conversion ALTER (http://www.sing-group.org/ALTER/) and KaKs_Calculator 2.0 soft-ware were used to calculate the nonsynonymous substitution rates (*Ka*), synonymous substitution rates (*Ks*), and *Ka/Ks* in the Linux operating environment.

### Candidate CYPs screening and cloning

CYPs were obtained from the PFAM database and annotated as cyto-chrome P450 (PF00067). A heatmap of gene expression was generated using MultiExperiment Viewer (MeV, version 4.9.0) and Java 8 soft-ware. The RPKM reads of the root periderm, root phloem, root xylem, stem vascular bundle, stem periderm, and leaf originated from pre-vious tissue transcriptomes (NCBI SRA number SRP199495, Supple-mentary Fig. 20)[16], and the data were normalized and processed by hierarchical cluster analysis with Pearson correlation. CYPs with gene expression profiles similar to those of *TwTPS7(v2)*, *TwTPS27(v2)*, and *CYP728B70* were selected as candidate CYPs for further analysis. The candidate CYPs were amplified using 2×Phusion High-Fidelity PCR Master Mix (New England Biolabs, USA) and cDNA of *T. wilfordii* root as the template, which was consistent with the transcriptome after complete sequencing (Supplementary Data 5).

### Heterologous expression and functional characterization in yeast

Candidate CYPs were integrated into the high-copy plasmid pESC-LEU expressing TwPOR3 from the GAL1 promoter and CYP from the GAL10 promoter (Supplementary Data 5). The recombinant plasmids were transformed into the yeast strain BY4741, and pESC-LEU with *TwPOR3* was employed as an empty vector control. For the in vivo assay, SD medium without tyrosine and leucine (SD-Trp-Leu) and 2% Glc was used to select positive strains, and 2% Gal was added to 20 mL of SD-Trp-Leu for the fermentation of fresh cells in a shaker at 30 °C and 200 rpm for 12 h. After the feeding of **7** or **8** to a final concentration of 50 μM, the resulting mixture was fermented for another 48-60 h, and ultrasonic extraction of fermentation products with an equal volume of ethyl acetate was then performed twice for 1 h each time. For the in vitro assay, microsomes were extracted according to the following experimental procedure. The base buffer TE (pH 7.5) consisted of 50 mM Tris-HCl and 1 mM EDTA. Cells were collected by centrifugation at 4000 × *g* for 3 min, resuspended in TEK (0.1 M KCl in TE), and left at room temperature for 10 min. The cells were collected again by cen-trifugation and resuspended in pre-cooled TESB (0.6 M sorbitol in TE). The cells were completely crushed using a cryogenic homogenizer (ATS, Canada) for 3–5 cycles. After centrifugation at 12,000 × *g* for 15 min, the microsomes were precipitated by adding NaCl at a final concentration of 0.15 M and polyethylene glycol PEG4000 at a final concentration of 0.1 g·mL$^{-1}$ to the supernatant. The pellets were resuspended in TEG (20% (v/v) glycerol in TE) for preservation and catalytic reactions[32]. The enzymatic assay was performed in a 500-μL system containing 100 mM Tris-HCl (pH 7.5), 1 mM nicotinamide ade-nine dinucleotide phosphate (NADPH), 4 mM glucose-6-phosphate, 1 unit of glucose-6-phosphate dehydrogenase, 5 μM flavin mono-nucleotide, 5 μM flavin adenine dinucleotide, 1 mM dithiothreitol, 0.5 mg of microsomal protein, and 100 μM substrate, which was incubated for 20 h at 30 °C with shaking at 100 rpm in the dark and then extracted with 500 μL of ethyl acetate three times. The organic phases of both the in vivo and in vitro samples were dried, recon-stituted with methanol and then derivatized with trimethylsilyl dia-zomethane (Tokyo Chemical Industry, Japan)[52]. The methylated samples were dissolved with ethyl acetate for GC-TQ-MS/MS analysis. For the **5**-catalyzed reaction, the plasmid was transformed into engi-neered yeast BY-PS2 (Supplementary Table 3).

### Enrichment, isolation, and NMR analysis of the product

Strains containing the pESC-LEU::(*CYP82D274* + *TwPOR3*) plasmid were inoculated into 500 mL of SD-Trp-Leu + 2% Glc fluid medium and grown at 30 °C and 200 rpm to an OD$_{600}$ of ~10. The cells were col-lected by centrifugation, evenly distributed into twenty 2-L flasks containing 500 mL of YPL and 100 μM substrate **7**, and then fermented at 30 °C and 200 rpm for 72 h. Then, 15-20 g of HP2MGL resin (Mit-subishi, Japan) was added to each flask in a sterile environment, and the fermentation product was extracted in a shaker at 200 rpm for another 24 h.

Ten liters of fermentation broth was centrifuged at 4000 × *g* for 5 min, and the yeast cells and resin were collected and dried in an oven at 40 °C. Eight times the volume of anhydrous methanol was added, and the catalytic product was extracted by ultrasonication at low temperature for 30 min. The solids were repeatedly extracted until the anhydrous methanol extract was colorless. The extracts were filtered to remove the resin and cell residues and evaporated under a vacuum at 40 °C. The products were dissolved with 4 mL of chromatography-grade methanol and sampled into a Gilson 281 semipreparative HPLC connected to an H322 pump, GX-281 autosampler, 156 dual wavelength UV detector, and Trilution LC 2.1 workstation. An Xtimate C18 column (30 mm × 75 mm × 3.0 μm, Welch) with 0.1% trifluoroacetic acid-water (mobile phase A) and acetonitrile (mobile phase B) was used. Gradient elution was started with 50% B and then increased to 100% B over

8.0 min, the flow rate was set to 25 mL·min⁻¹, and the injection volume was 500 µL. The product fractions were collected at 7.4–7.8 min and freeze-dried. Three milligrams of product powder was weighed precisely and fully dissolved in 500 µL of deuterated acetone ($C_3D_6O$, InnoChem, China). For chemical structure characterization, ¹H-NMR (800 M), ¹³C-NMR (200 M), NOESY, HSQC, and ¹H-¹H COSY were analyzed using a Bruker Avance III HD (Bruker BioSpin, Germany). MestReNova 14 software was used to analyze the data.

### Kinetic analysis

The CYP concentration was estimated by measuring the reduced carbon monoxide difference spectrum using an extinction coefficient (450 versus 490 nm) of 91 mM⁻¹·cm⁻¹[53]. Incubation times from 0.5 to 8 h were used. The $K_m$ values were determined using serial concentrations from 0.1 to 100 µM for **7** and 5 to 80 µM for **5** in 300-µL enzyme assays. These also contained ~0.4 mg of microsomal protein, Tris-HCl (pH 7.5), NADPH, and the regenerating system mentioned above. The reactions were initiated by substrate addition and incubated with shaking at 30 °C for 30 min in the dark. Extraction was then terminated three times with 300 µL of ethyl acetate, and the extracts were completely dried with a vacuum concentrator (Eppendorf Concentrator plus, Germany). The product **11** was quantified using 100 µL of methanol dissolved for LC-TQ-MS/MS analysis, and the product **14** was dissolved in 100 µL of ethyl acetate for quantification by GC–MS. The $K_m$ values were calculated by nonlinear regression using GraphPad Prism version 7.

### Gene overexpression and RNA interference in suspension cells

The specific fragments of *CYP82D274* and *CYP82D263* were chosen via multiple sequence alignment of all CYP82D nucleic acid sequences in *T. wilfordii*. The specific fragments (Supplementary Data 5) were inserted into the pH7WG2D (for OE) and pK7GWIWG2D (for RNAi) binary vectors. Using the developed gene gun technology[25,52], recombinant plasmids were transformed into suspension cells that had been precultured for 7 days and then cultured for another 7 days to collect samples. RNA extraction and reverse transcription were performed using Total RNA Extraction Kit (Promega, China) and FastQuant RT Kit (Tiangen Biotech, China)[51], and gene expression was detected using qRT–PCR (Supplementary Data 5). Twenty milligrams of freeze-dried sample powder was precisely weighed, soaked in 1 mL of 80% (v/v) methanol overnight at 4 °C, and subjected to ultrasonication at 40 kHz for 1 h at 25 °C. After centrifugation at 13,000 × g for 10 min, the supernatant was filtered through a 0.22-µm PTFE microporous membrane to obtain the samples for Q Exactive HF LC–MS/MS.

### Metabolite feeding studies

The substrate feeding was performed by soaking 0.2 g of suspension cells with 1 mL of MS medium containing 0.8 mM **11** for 7 days before sampling, and the negative control was soaked using an equal volume of MS liquid. For stable isotope labeling assay, ¹⁸O-**11** was obtained by expanding the above microsomal assay system to 600 mL and completely replacing ¹⁶O₂ with ¹⁸O₂ (purity > 97%) before the addition of substrate **7**, and after 1 h of reaction, an equal volume of ethyl acetate was extracted three times and then subjected to product enrichment and isolation. The purified ¹⁸O-**11** was determined by LC-qTOF-MS. After gene gun bombardment of *TwTPS7*&*TwTPS27*-RNAi (Supplementary Data 5) for 5 days, ~0.2 g of cells were soaked in 1 mL of MS medium with 0.4 mM ¹⁸O-**11** and 100 µM fosmidomycin for 5 days and then sampled after 2 days of induction with 50 µM methyl jasmonate. Equal concentrations of unlabeled **11** with identical treatment and WT cells were used as controls. Three biological replicates of each group were included in the experiment. The metabolite extraction was performed as described above, with Q Exactive HF LC–MS/MS for quantitative assays and LC-qTOF-MS for qualitative assays.

### Yeast engineering

For CYP coexpression, the recombinant plasmids pESC-LEU::(*CYP82Ds* + *TwPOR3*) and pESC-TRP:: (*CYP728B70* + *TwPOR3*) were chemically transformed into the yeast strain BY-PS2 together or separately. A corresponding auxotrophic medium was employed for selection and fermentation, and products were extracted with ethyl acetate and detected via GC–MS. For the screening of CYP and POR combinations, the recombinant plasmid pYES2:: (P$_{TDH3}$-*tSmKSL1*-*CfTPS1*-T$_{TPII}$-P$_{ADHI}$-*CYP82D274*-T$_{PGI}$) was constructed (Supplementary Data 5) using BY-HZ16 strain homologous recombination to generate the BY-ZY1 strain. Truncated transit peptides (50 amino acids at the N-terminus predicted by TargetP Server v2.0) or complete CYP728B70 was constructed at multiple cloning site (MCS) 1, and a truncated transmembrane domain (67 amino acids for TwPOR1 and 46 amino acids for TwPOR3 and TwPOR4, predicted by TMHMM Server v2.0) or complete TwPOR was integrated at MCS2 of pESC-LEU (Supplementary Data 5). Twelve enzyme combinations (Supplementary Table 2) were transformed into the BY-ZY1 strain. For optimization of the chassis strain, the haploid strains BY4741H (Supplementary Table 3) and BY-HZ16 were inoculated in the respective media and incubated at 30 °C and 200 rpm for 24 h. The two strains were mixed in SD-His-Trp supplemented with 2% Glc liquid medium at an equal OD and cocultured for another 24 h. After centrifugation, the cells were collected and coated on SD-His-Trp with 2% Glc solid plates and incubated at 30 °C for 72 h, and the monoclonal strains were selected. The newly constructed diploid strain was identified and named BY-ZY2D. The *tSmKSL1-CfTPS1* and *CYP82D274* (or mutants) modules were cloned and integrated into the haploid strain BY-HZ16 and diploid strain BY-ZY2D chromosome *YPRCΔ15* locus by the modularized two-step chromosome integration technique[54] using *Ura3* as a selection marker (Supplementary Fig. 21 and Supplementary Data 5). Subsequently, pESC-LEU::(*CYP720B4* + *TwPOR3*) was chemically transformed to generate BY-ZY11H to ZY14H (haploid strains) and BY-ZY15D to ZY18D (diploid strains) (Supplementary Table 3). For quantitative inoculation, the seed strain was transferred into 20 mL of fresh medium with a quantitative OD of 0.1, and three biological replicates of each group were included in the experiment. The product was extracted with an equal volume of n-hexane and dissolved in 100 µL of methanol for LC-TQ-MS/MS.

### MALDI-MSI

Matrix-assisted laser desorption/ionization-mass spectrometry imaging (MALDI-MSI) is a label-free technique used for tissue sample imaging that allows the localization of known or unknown biological molecules in a spatial region within subcellular compartments. The sample is rasterized while a laser is used as a stationary ionization source, and the mass spectrum is utilized for recording and converting the ion signal intensity and then drawing the XY coordinate, enabling the spatial distribution of the ion of interest for visualization[55,56]. High-coverage MALDI-MSI was employed to map the target molecules in the root of *T. wilfordii*. The fresh root was flash-frozen in liquid nitrogen and then maintained at −80 °C for over 8 h. The root tissue was cryosectioned into 20-µm sections on a cryostat microtome (CM3050S, Leica Biosystems, Germany) in a −20 °C environment and attached to the conductive side of an indium tin oxide (ITO)-coated glass slide (Bruker, Germany). The glass slide was kept in a vacuum desiccator and dried for 30 min. Then, 2,5-dihydroxybenzoic acid (DHB) was selected as the matrix under the positive ion mode after the pre-experiment and was sprayed evenly on the surface of the slide by an HTX™-Sprayer (HTX Technologies, USA). MALDI-MSI was conducted with a timsTOF fleX™ mass spectrometer (Bruker), and the parameter settings were as follows. The laser frequency and accumulated shots were set to 10,000 Hz and 500 shots, respectively, the spatial resolution was 50 µm, and the target profile was 18.70 µm. The raw mass spectra data were acquired over the *m/z* range of 200-1000. A Bruker SCiLS Lab

2020a instrument and MetaboScape software were used to analyze the MALDI imaging data and visualize the *m/z* values.

## GC-TQ-MS/MS

GC-TQ-MS/MS was performed on an Agilent 7890B GC system equipped with a 7000 C GC/MS Triple Quad. For the DB-5MS (15 m × 0.25 mm × 0.1 μm) capillary column, splitless injection was performed at 250 °C. The GC oven temperature was adjusted according to the following program: an initial temperature of 50 °C was maintained for 1 min, followed by an increase to 200 °C at a rate of 40 °C·min⁻¹, an increase to 240 °C at a rate of 20 °C·min⁻¹, an increase to 246 °C at a rate of 1.5 °C·min⁻¹, and an increase to 300 °C at a rate of 40 °C·min⁻¹, with the final temperature being maintained for 3 min. For the DB-5MS (30 m × 0.25 mm × 0.1 μm) column, the oven temperature program was as follows: 50 °C isothermal for 1 min, increased to 240 °C at 50 °C·min⁻¹, increased to 255 °C at 1.5 °C·min⁻¹, increased to 300 °C at 50 °C·min⁻¹ and maintained at the final temperature for 1 min. Helium was used as the carrier gas at a flow rate of 1 mL·min⁻¹. The inlet temperature was set to 300 °C, the ion trap temperature was 250 °C, the electron energy was 70 eV, and the spectra were recorded over a range of 10-400 *m/z*. Qualitative Analysis software (B.07.00) was used for data analysis.

## LC-TQ-MS/MS

LC-TQ-MS/MS was performed with an Agilent 1290 Infinity LC tandem QTRAP 6500 MS (AB SCIEX). We used an ACQUITY UPLC HSS T3 column (2.1 mm × 100 mm × 1.8 μm, Waters) with 0.1% formic acid-water as mobile phase A and acetonitrile as mobile phase B. A flow rate of 0.3 mL·min⁻¹ was used, and the gradient program was as follows: 0–1 min, 40% B; 2–3 min, 90% B; and 5–6 min, 40% B. In each run, 3 μL of the sample was analyzed, and the column temperature was 40 °C. The quantitative ions for multiple reaction monitoring (MRM) **11** were 315 and 269 *m/z*.

## Q Exactive HF LC–MS/MS

Q Exactive HF LC–MS/MS (Thermo Scientific) was performed on an ACQUITY UPLC HSS T3 column (2.1 mm × 100 mm × 1.8 μm, Waters) with 0.1% formic acid-water as mobile phase A and acetonitrile as mobile phase B. A flow rate of 0.4 mL·min⁻¹ was used, and the gradient program was as follows: 0–4 min, 30–34% B; 4–8 min, 34–52% B; 8–12 min, 52% B; 12–23 min, 52–77% B; and 23–26 min, 90% B. In each run, 2 μL of the sample was analyzed, and the column temperature was 40 °C. When the parent ion quantification method was applied, either positive or negative ion mode was used for different metabolites.

## LC-qTOF-MS

LC-qTOF-MS (Waters) was performed on an ACQUITY UPLC HSS T3 column (2.1 mm × 100 mm × 1.8 μm, Waters) with 0.1% formic acid-water as mobile phase A and acetonitrile as mobile phase B. The gradient program was as follows: 0–0.5 min, 20% B; 0.5–21 min, 20–85% B; 21–23 min, 85–100% B; 23–26 min, 100% B; 26–27 min, linear decrease from 100 to 20% B; and 27–30 min, isocratic 20% B. The flow rate was set at 0.4 mL·min⁻¹. Mass spectra were acquired in positive/negative ion mode over a scan range of *m/z* 50-1200 with a scan time of 0.2 s. The MS settings were as follows: resolution analyzer mode; normal dynamic range; ramp collision energy 10-50 V; cone voltage 40 V. MassLynx v4.2 was used for data analysis.

## qRT–PCR

Primers (Supplementary Data 5), cDNA, and reagents were added according to the TransStart Top Green qPCR SuperMix (TransGen Biotech) instructions. QuantStudio5 (Applied Biosystems) was employed to assay the expression of genes, and each reaction was repeated three times. The mixed reagents were first activated for 30 s at 94 °C, and then denaturation occurred for 5 s at 94 °C, followed by

annealing for 15 s at 56 °C and extension for 10 s at 72 °C. This procedure was repeated for 45 cycles, followed by the dissociation stage. *EFLα* (Supplementary Data 5) was used as the housekeeping reference gene. The relative expression of genes was analyzed using the $2^{-\Delta\Delta Ct}$ method[57].

## Reporting summary

Further information on research design is available in the Nature Portfolio Reporting Summary linked to this article.

## Data availability

The genome and transcriptome sequence data are available at NCBI BioProject PRJNA542587. Full sequences of CYP82D274 (accession XP_038717985) and CYP82D263 (accession XP_038716720) are deposited in GenBank. The databases of TargetP v2.0 (https://services.healthtech.dtu.dk/service.php?TargetP-2.0) and TMHMM v2.0 (https://services.healthtech.dtu.dk/service.php?TMHMM-2.0) are used for data analyses. Source data are provided with this paper.

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

## Acknowledgements

The authors thank Prof. Juan Guo from the National Resource Center for Chinese Materia Medica for writing and experimental guidance. We thank Dr. Pengfei Li and Kairong Wu from Bruker (Beijing) Scientific Technology Co., Ltd for MALDI-MSI. We appreciate the guidance of Dr. Yaqiu Zhao from National Resource Center for Chinese Materia Medica in the isotopically labeled compound. We thank Dr. David Nelson from the University of Tennessee for assigning CYP nomenclature. We thank Prof.

Fei Li from the Kunming Institute of Botany for providing chemical standard. This work was supported by the National Key R&D Program of China (2020YFA0908000) to L.H. and W.G., the Innovation Team and Talents Cultivation Program of National Administration of Traditional Chinese Medicine (ZYYCXTD-D-202005) to W.G., the National Natural Science Foundation of China (82204547), the Key Project at central government level: The ability establishment of sustainable use for valuable Chinese medicine resources (2060302), and the Fundamental Research Funds for the Central public welfare research institutes (ZZXT202206) to W.G. and Yi.Z.

## Author contributions

Yi.Z., W.G., and L.H. conceived and initiated the study. Yi.Z., J.G., L.M., Y.Liu, J.Z., Y.Lu, and J.W. performed the experiments. L.T. performed the phylogenetic analysis. Yu.Z., Y.T., and H.Z. provided the resources and materials. Yi.Z., X.W., and P.S. analyzed the quantitative data. D.L. conducted the NMR analysis. T.H., and Y.Y. established yeast strains used. Yi.Z. and W.G. wrote the manuscript. W.G. and L.H. revised the manuscript.

## Competing interests

Yi.Z., J.G., L.M., and W.G. are inventors of Chinese invention patents (application No. CN202210059108.2 and CN202210059174.X) related to the cytochrome P450 enzymes described in the paper. Other authors declare no competing interests.
