## [Peer Review File · Nature Communications]

Tandemly duplicated CYP82Ds catalyze 14-hydroxylation in triptolide biosynthesis and precursor production in *Saccharomyces cerevisiae*Reviewers' Comments:

Reviewer #1:

Remarks to the Author:

I make a comment only on MALDI imaging part. Although not definitive, MS imaging data support the author's claim, particularly considering relatively accurate mass resolution and co-localization of the compounds in the synthetic pathway. The signal for #11 might have been more pronounced if the data were collected in negative mode due to the deprotonation of carboxylic acid, further supporting the claim; however, it may not be super necessary. Other parts of the manuscript seem all reasonable, although they are beyond my expertise.

Reviewer #2:

Remarks to the Author:

Triptolide is one of the bioactive natural products produced by medicinal plant *Tripterygium wilfordii*. The unique epoxide and lactone structural features of triptolide pose interesting questions regarding its biosynthetic mechanism. In this manuscript, the authors took a transcriptome-based candidate gene approach, and identified two CYP82-family P450 enzymes, CYP82D274 and CYP82D263, which potentially participate in triptolide biosynthesis in *Tripterygium wilfordii*. Using recombinant expression in yeast, the authors report that CYP82D274 and CYP82D263 can catalyze C-14 hydroxylation of compound 7 and aromatization of compound 5 to generate compound 6 (numbered compounds as proposed triptolide biosynthetic intermediates). Based on enzyme assays carried out using recombinant yeast system, the authors concluded that CYP82D274 participates in triptolide biosynthesis in the context of a metabolic network. Evolutionary analyses of the origin of these CYP82D paralogs in *Tripterygium wilfordii* were also pursued.

Overall, this work is relatively narrow in scope and is preliminary in nature. The data presented on the biochemical functions of the two CYP82D members were solely based on enzyme assays carried out in transgenic yeast or using in vitro enzyme assays framed under the proposed triptolide biosynthetic pathway. No kinetic data were presented, making it difficult to evaluate whether these detected activities correspond to the in vivo biochemical functions of these enzymes. The conclusion that CYP82D274 participates in triptolide biosynthesis in the context of a metabolic network is not sufficiently supported by the data presented. Other supporting evidences, such as in planta reconstitution, VIGS, isotopic tracing, and metabolite profiling in host plant, would help clarify the triptolide biosynthetic pathway and the role of these enzymes in vivo. Even with that, the overall impact of the study is still limited, as the described biochemistry (i.e., hydroxylation and oxidative aromatization) is predicted chemistry catalyzed by P450, and the most interesting questions regarding triptolide biosynthesis remain unsolved.

There are also many problems with how this manuscript is written, which I'm detailing below.

- The title "Tandem duplication of CYP82Ds catalyzes the 14-hydroxylation metabolic network in triptolide biosynthesis and precursor production in *Saccharomyces cerevisiae*" is problematic because tandem duplication is not a proper noun for catalyze. "Catalyze metabolic network" is also incorrect.
- Line 10, "characterize CYP82D274 and CYP82D263 as 14-hydroxylase catalyzing intermediate dehydroabiatic acid, ". Catalyzing an intermediate is incorrect.
- Line 18, "as well as broad insights into the current intractable biosynthesis of secondary metabolites." This is too vague and does not conduct useful information specific to this study.
- Line 28, "Despite this rich compound treasury, triptolide (1) is still an important contributor to pharmaceutical properties." Please site supporting literature here. The use of "despite" here is not appropriate.
- Line 33, "and anti-osteoporosis activities, which can significantly inhibit lung cancer, liver cancer, pancreatic cancer, rheumatoid arthritis, systemic lupus erythematosus, leukemia, Parkinson's disease, etc." Since triptolide is not an approved pharmaceutical for any disease, this statement is unsubstantiated and misleading.

- Line 41, Should be structural analogs instead of "Dozens of structural-based derivatives".
- Line 59, "Exclusive triptolide biosynthesis is" is incorrect. You are referring to the first committed step.
- Line 67-73, the mention of medicinal activities here is out of place.
- Line 73-75 and Line 90-91 are redundant with what's said in Line 56-58.
- Line 86, "located on the length of the 133 kb tandem duplication", check grammar for the sentence.
- Line 302, the author did not describe the nature of the transcriptome resource and how it was explored for co-expression analysis in the results section, making it hard to comprehend how candidate genes were identified.
- Line 309-312, I don't know what this sentence means.
- Line 321 and other places, Emotional expression, such as "gratifyingly" or "surprisingly" should not be used in a scientific paper.
- Line 329-330, compounds cannot be faithfully IDed simply based on mass spectrum matching.
- Line 338, I found the mutational analysis section lacking rationale. What is the question or questions that motivated these studies? What were learnt about the roles of these residues from the results? I also feel the exploration of residue functions should follow evolutionary analysis of CYP82D paralogs. Are there specialization mutations to the two enzymes discovered in this study relative to other CYP82 enzymes?
- Line 414, "CYP82D274 or CYP82D263 coexpressed with CYP728B70 in yeast" is not a google section title. Try to use a phrase or single declarative sentence to summarize what's learnt.
- Line 415 and 434, Please use alternative terms to describe "generation modules". It's confusing to readers.
- Line 500, when interpreting phylogenetic analysis and other evolutionary analyses, it is important to recognize these are inferred possibilities instead of absolute truth.
- Line 523 and 530, genes "were not transcribed" is incorrect. Should say something like "the expression of these genes was too low to be detected".
- Line 553, The MALDI data should be presented in full in the results section instead of in the discussion.
- I find many figure legends lacking sufficient information necessary for the readers to understand these figures.

Reviewer #3:

Remarks to the Author:

The authors have generated some really interesting data, and shown further evidence that CYPs play a huge role in plant specialized metabolism. The work is significant and very promising and is based on well established models.

Overall the paper is fine, though I have some problems that needs to be addressed.

The introduction is very lengthy and provides way to much information. You do not have to justify why to work with triptolide. This is already a well known compound. Thus I suggest to delete the following paragraph (almost 2 pages).

Line 25-29; Line 32-44 (then delete however); line 52-56; line 68-75. All this work is redundant.

The authors should concentrate the introduction on what is discussed. Also I miss some introduction on other CYP's involved in diterpenoid biosynthesis, e.g. forskolin (also made in yeast).

Cytochrome P450 reductases have now for about 10 years been abbreviated POR and NOT CPR like the authors have chosen. Even wikipedia is updated on this. I will require that the authors fix this through out. CPR is used for other enzymes, thus the change.

Specific comments on the results.

Minor thing, but in headings, please use full name of chemical structures e.g. line 380

In line 420-423 it states that CYP82D270 co-expressed with C263 gives higher yield than D270 alone. I cannot find those data - I think this is a crucial point and should be discussed further in connection with the speculative substrate of D263 being 6 and not 7. Here again the authors should consult the work on Forskolin, where similar observations was made.

Line 423 "the cyp728D70 product" - use the number 7 instead of this odd naming.

Line 429 - again networks/grid are also seen e.g. in forskolin. THIS is a link to just one of the papers about this compound. <https://elifesciences.org/articles/23001>

Line 440-465 - PORs are normally divided into two clades POR1 and POR2 (<https://www.sciencedirect.com/science/article/pii/S0031942209004567>) - normally it is observed that the inducible PORs have stronger electron transfer than the constitutive. How does the PORs from *T. wilfordii* fall into these two clades - and please annotate them 1a or 2a respectively of their clade. You can also see more here:

<https://www.sciencedirect.com/science/article/pii/S1055790316000294#b0100>

Else the POR findings are very interesting and would be nice if this little extra biological information could be added.

Line 458, add paragraph change before the picia discussion, please.

As for the strain optimization - consult this thesis for inspiration to the discussion - cannot remember which paper it is in, but the thesis discuss the issue on strain.

https://backend.orbit.dtu.dk/ws/portalfiles/portal/242032750/PhD_thesis_SEBRTH.pdf

Not sure how the chromosome localization add to this paper. I will suggest to take this part out, and publish by it self. Maybe in Molecular Phylogenetics and Evolution. I really struggle with finding the connection to the rest of the story for this part.

Good work, and look forward to see it "in print" ;)

As for the discussion of 82D as family. Is the conclusion not that 82D hydroxylate aromatic multiple ring compounds and that the sub-family can be divided into flavonoid clade, coumarin, terpenoid etc - I would love to see the tree (sup figure 19) as part of the main figures - rather than the chromosome discussion.

Reviewer #4:

Remarks to the Author:

In the manuscript, the authors reported the discovery of two new CYP450 enzymes which were responsible for the 14-hydroxylation of multiple triptolide intermediates. They first analyzed transcriptional data for *T. wilfordii*, triptolide producer, to identify genes co-expressed with other known triptolide pathway genes, which resulted in the discovery of two CYPs that can catalyze the 14-hydroxylation. Then, they tried to analyze these two genes by in vivo and in vitro experiments with feeding of different precursors as well as mutagenesis studies to explore their mechanism, during which they also found the enzymes can catalyze one step previously known as spontaneous reaction. The authors then tried to perform metabolic engineering to achieve the de novo biosynthesis of 14-hydroxy-dehydroabiatic acid in yeast, although the exploration was simple probably due to the page limit. Last, the author tried to study the evolution of these genes through chromosome localization analysis and formed a hypothesis with limited clues. Overall, the discovery of these genes is important, which brings the whole community closer to the complete biosynthesis of triptolide. However, the organization of this manuscript and the description of experiments need a lot of improvements before it can be further considered for publication.

Major points:

1. The authors tried to study the two enzymes from many different perspectives (mechanistic, structural, evolutionary, biosynthesis, etc.). Although it is good for the audience to know more about the target enzymes, the page/word limit prevent the author from describing all the information in details. It would be good for the authors to narrow the scope of this manuscript, focusing either on the discovery of enzymes and establishment of total biosynthetic pathways, or on the detailed characterization of the evolution/mechanism of these enzymes.
2. Similar to point 1, the authors even had original research in their discussion section, which is another sign of lack of space for the overwhelming content.
3. The authors need improve their writing/figure quality to describe their work clearer, both scientifically and language-wise.

a) Scientific issues:

Line 318: TwCPR3, when first introduce an enzyme, it is important to have a full name for them. Many other genes have the same issue (Line 435,449, etc).

Line 327: which partner CPR?

Line 383: "we transformed plasmids" – which plasmids? Into which engineered yeast?

Line 384: "self-produced" – does it mean self-produced by yeast, or in-house production?

Line 449: How were they truncated?

Line 476: which genes were integrated? Which were on the plasmids?

b) Language issues:

Line81-82: "via a metabolic network" is difficult to understand. It seems that the authors tried to mean that triptolide biosynthesis is not linear and some specific intermediates can be synthesized from the same precursor through different pathways. If true, the sentence is no clear.

Line86: "located on the length of the 133 kb" is not understandable.

Line297: "share" the biosynthesis through coexpression is not clear.

Line339-350: This paragraph is in isolation, the authors need to integrate it with the next paragraph.

Line366: "even though the A326S ... products disappeared" is not a clear sentence.

Line 415: "three generation modules" is not clear

c) Figure issues:

Fig1C: difficult to draw the conclusion of "same pattern" from the figure.

Fig2b: It should be proposed catalytic reaction.

FigS10, Fig3f and more: it would be better to label every sub-spectra.

Fig4a: whether all spectrum have the same y-axis? It seems not.

Too many to point out one by one, the authors need to have a careful revision of the manuscript.

Minor points:

1. Better to have atom number of one of the molecules to help the audience understand all the positions.
2. Line 459: why not select CYP720B4 first?

Response to reviewers

Dear Reviews,

We are very grateful for your valuable comments and suggestions which improve the quality of this manuscript. We have carefully taken your comments into consideration in preparing our revision and have made modifications and corrections. Here below is our description of revision, and we sincerely hope it meets with your approval. We wish you a Merry Christmas and Happy New Year at this wonderful moment!

Reviewer #1

Reviewer 1 Comment 1

I make a comment only on MALDI imaging part. Although not definitive, MS imaging data support the author's claim, particularly considering relatively accurate mass resolution and co-localization of the compounds in the synthetic pathway. The signal for #11 might have been more pronounced if the data were collected in negative mode due to the deprotonation of carboxylic acid, further supporting the claim; however, it may not be super necessary. Other parts of the manuscript seem all reasonable, although they are beyond my expertise.

Response: Thank you for your review of our manuscript and suggestion. We employed MALDI-MSI with 2,5-dihydroxybenzoic acid (DHB) under positive ion mode to confirm the reasonableness of the selection of intermediates in the proposed biosynthetic pathway of triptolide. As we have seen, several intermediates show a tendency to be localized in the periderm distribution. However, we also tried the negative mode of the DHB matrix in pre-experiments, with insignificant MSI. The results are as follows (Fig. R1). That's why we chose the positive mode and hope to get your approval.

Fig. R1 MALDI assay for the distribution of triptolide intermediates in *Tripterygium wilfordii* root tissue with negative mode.

Reviewer #2

Reviewer 2 Comment 1

Triptolide is one of the bioactive natural products produced by medicinal plant *Tripterygium wilfordii*. The unique epoxide and lactone structural features of triptolide pose interesting questions regarding its biosynthetic mechanism. In this manuscript, the authors took a transcriptome-based candidate gene approach, and identified two CYP82-family P450 enzymes, CYP82D274 and CYP82D263, which potentially participate in triptolide biosynthesis in *Tripterygium wilfordii*. Using recombinant expression in yeast, the authors report that CYP82D274 and CYP82D263 can catalyze C-14 hydroxylation of compound 7 and aromatization of compound 5 to generate compound 6 (numbered compounds as proposed triptolide biosynthetic intermediates). Based on enzyme assays carried out using recombinant yeast system, the authors concluded that CYP82D274 participates in triptolide biosynthesis in the context of a metabolic network. Evolutionary analyses of the origin of these CYP82D paralogs in *Tripterygium wilfordii* were also pursued.

Overall, this work is relatively narrow in scope and is preliminary in nature. The data presented on the biochemical functions of the two CYP82D members were solely based on enzyme assays carried out in transgenic yeast or using in vitro enzyme assays framed under the proposed triptolide biosynthetic pathway. No kinetic data were presented, making it difficult to evaluate whether these detected activities correspond to the in vivo biochemical functions of these enzymes. The conclusion that CYP82D274 participates in triptolide biosynthesis in the context of a metabolic network is not sufficiently supported by the data presented. Other supporting evidences, such as in planta reconstitution, VIGS, isotopic tracing, and metabolite profiling in host plant, would help clarify the triptolide biosynthetic pathway and the role of these enzymes in vivo. Even with that, the overall impact of the study is still limited, as the described biochemistry (i.e., hydroxylation and oxidative aromatization) is predicted chemistry catalyzed by P450, and the most interesting questions regarding triptolide biosynthesis remain unsolved.

There are also many problems with how this manuscript is written, which I'm detailing below.

Response: Thank you for your patient review and guidance of our manuscript, and we have complemented the studies to verify CYP82D274 and CYP82D263 participate in triptolide biosynthesis through MALDI, metabolite feeding suspension cells, gene overexpression and RNA interference (Fig. 4). Hope to get your approval.

1. MALDI.

The 14-hydroxy-dehydroabietic acid (**11**) accumulates in the periderm range (Fig. 4a), which is similar with the distribution pattern of the other acquainted intermediates abietatriene (**6**), dehydroabietic acid (**7**) and triptophenolide (**10**). -- Demonstrate **11** as a potential intermediate from the plant tissue level.

2. Metabolite feeding suspension cells

We have fed **11** to *T. wilfordii* suspension cell cultures and found that the content of

triptolide (**1**) and **10** was elevated 35.73 and 35.60 times, respectively, compared with the control (Fig. 4b and Supplementary Data 2). Meanwhile, the catalytic substrate 7 of CYP82D274 and CYP82D263 was sharply reduced. We also detected the expression of biosynthetic pathway genes. -- Direct confirmation that **11** is a precursor of **1**.

3. CYP82D274 and CYP82D263 overexpression in suspension cells

-Since *Tripterygium wilfordii* is a woody vine, we have tried various methods to verify the *in vivo* functions of genes, induction of hairy root and VIGS failed to achieve. We finally used the biotechnology of genegun bombardment ¹ and successfully verified the biochemical functions of CYP728B70 ² and terpenoid upstream genes ^{3,4} *in vivo*.

- Metabolite profiling in CYP82D274 and CYP82D263 overexpressing cell lines. The content of **1** and intermediates **10** and **11** increased 1.36 ($P<0.001$), 1.60 ($P<0.05$) and 1.39-fold ($P<0.05$) when CYP82D274 was overexpressed, and enhanced 1.68 ($P<0.05$), 1.57 ($P<0.05$) and 1.36-fold ($P<0.01$), respectively, under the condition of CYP82D263 overexpression (Fig. 4d and Supplementary Data 3). --Direct confirmation the involvement of CYP82D274 and CYP82D263 in **1** biosynthesis.

-Notably, the content of intermediate 8 decreased to 0.69-fold ($P<0.01$) when CYP82D274 was overexpressed. --Leads to further exploration of other catalytic processes in CYP82D274 and CYP82D263 genes.

4. RNAi

The results showed that the content of final product **1** decreased to 0.38-fold ($P<0.01$) at CYP82D274-RNAi and 0.59-fold ($P<0.05$) at CYP82D263-RNAi (Fig 4e and Supplementary Data 4). --Direct confirmation the involvement of CYP82D274 and CYP82D263 in **1** biosynthesis.

Refs:

1. Zhao, Y. Zhang, Y. Su, P. Yang, J. Huang, L. & Gao, W. Genetic transformation system for woody plant *Tripterygium wilfordii* and its application to product natural celastrol. *Front. Plant Sci.* **8**, 2221 (2017).
2. Tu, L. et al. Genome of *Tripterygium wilfordii* and identification of cytochrome P450 involved in triptolide biosynthesis. *Nat. Commun.* **11**, 971 (2020).
3. Zhang, Y. et al. The expression of *TwDXS* in the MEP pathway specifically affects the accumulation of triptolide. *Physiol. Plant* **169**, 40-48 (2020).
4. Zhang, Y. et al. Overexpression and RNA interference of *TwDXR* regulate the accumulation of terpenoid active ingredients in *Tripterygium wilfordii*. *Biotechnol. Lett.* **40**, 419-425 (2018).

Reviewer 2 Comment 2

- The title “Tandem duplication of CYP82Ds catalyzes the 14-hydroxylation metabolic network in triptolide biosynthesis and precursor production in *Saccharomyces cerevisiae*” is problematic because tandem duplication is not a proper noun for catalyze. “Catalyze metabolic network” is also incorrect.

Response: The title has been revised to “The tandem duplicated CYP82Ds catalyze the 14-hydroxylation in triptolide biosynthesis and precursor production in

Saccharomyces cerevisiae".

Reviewer 2 Comment 3

- Line 10, "characterize CYP82D274 and CYP82D263 as 14-hydroxylase catalyzing intermediate dehydroabietic acid,". Catalyzing an intermediate is incorrect.

Response: Thank you for your patient guidance and apologize for our inappropriate statement. We have revised the statements in the Abstract.

Reviewer 2 Comment 4

- Line 18, "as well as broad insights into the current intractable biosynthesis of secondary metabolites." This is too vague and does not conduct useful information specific to this study.

Response: We have revised to "Our study provides important genetic elements for further elucidation of downstream biosynthetic pathways and synthetic biology production of triptolide, as well as the current intractable biosynthesis of other 14-hydroxyl abietane-type secondary metabolites."

Reviewer 2 Comment 5

- Line 28, "Despite this rich compound treasury, triptolide (1) is still an important contributor to pharmaceutical properties." Please site supporting literature here. The use of "despite" here is not appropriate.

Response: We have added supporting literature and revised to "Among this rich compound treasury, triptolide (1) is undoubtedly an important contributor to pharmaceutical properties".

Reviewer 2 Comment 6

- Line 33, "and anti-osteoporosis activities, which can significantly inhibit lung cancer, liver cancer, pancreatic cancer, rheumatoid arthritis, systemic lupus erythematosus, leukemia, Parkinson's disease, etc." Since triptolide is not an approved pharmaceutical for any disease, this statement is unsubstantiated and misleading.

Response: Thank you for pointing out our mistake! As you stated, triptolide is not an approved pharmaceutical whose potent bioactivities were demonstrated by *in vitro* assays or animal models. We have revised the incorrect statements.

Reviewer 2 Comment 7

- Line 41, Should be structural analogs instead of “Dozens of structural-based derivatives”.

Response: The “derivative” refers to the new compound obtained by artificial modification based on the structure. So, we have kept the statement of derivative after consulting the references and have adjusted the manuscript appropriately.

Reviewer 2 Comment 8

- Line 59, “Exclusive triptolide biosynthesis is” is incorrect. You are referring to the first committed step.
- Line 67-73, the mention of medicinal activities here is out of place.
- Line 73-75 and Line 90-91 are redundant with what’s said in Line 56-58.
- Line 86, “located on the length of the 133 kb tandem duplication”, check grammar for the sentence.

Response: We have modified the sentence to “the core skeleton cyclization and functional decoration of triptolide biosynthesis is initiated by...”, and have deleted and modified the redundant or grammatically incorrect sentences.

Reviewer 2 Comment 9

- Line 302, the author did not describe the nature of the transcriptome resource and how it was explored for co-expression analysis in the results section, making it hard to comprehend how candidate genes were identified.
- Line 309-312, I don’t know what this sentence means.
- Line 321 and other places, Emotional expression, such as “gratifyingly” or “surprisingly” should not be used in a scientific paper.
- Line 329-330, compounds cannot be faithfully IDed simply based on mass spectrum matching.

Response:

- Line 302. We have made additional descriptions.
- Line 309-312. We wanted to express that CYP82 had been reported to be involved in terpenoid biosynthesis, so we speculated that CYP82D might also catalyze terpenoids. We have corrected the sentences that were difficult to understand, thank you for pointing out our problem.
- Line 321. Sorry for the emotional expressions, we have deleted them.
- Line 329-330. The structure of **13** was determined by comparison with authentic standard 15-hydroxy-dehydroabiatic acid (Supplementary Fig. S8).

Reviewer 2 Comment 10

- Line 338, I found the mutational analysis section lacking rationale. What is the question or questions that motivated these studies? What were learnt about the roles of these residues from the results? I also feel the exploration of residue functions should follow evolutionary analysis of CYP82D paralogs. Are there specialization mutations to the two enzymes discovered in this study relative to other CYP82 enzymes?

Response: Thank you for your guidance. As you said this section is missing the rationale and does not fit with the whole story about “the discovery of enzymes and establishment of total biosynthetic pathways”. This view is also consistent with the suggestions of Reviewer 3 and Reviewer 4. We have therefore removed “mutational analysis” section. We will expand our mutational research as you suggest, thanks again!

Reviewer 2 Comment 11

- Line 414, “CYP82D274 or CYP82D263 coexpressed with CYP728B70 in yeast” is not a google section title. Try to use a phrase or single declarative sentence to summarize what’s learnt.
- Line 415 and 434, Please use alternative terms to describe “generation modules”. It’s confusing to readers.

Response: Thank you for pointing out our problem. After careful consideration, we have modified the titles in this section and the next section to “*De novo* biosynthesis of 14-hydroxy-dehydroabietic acid in yeast” and “Synthetic biology strategies to enhance the yield of 14-hydroxy-dehydroabietic acid”.

Reviewer 2 Comment 12

- Line 415 and 434, Please use alternative terms to describe “generation modules”. It’s confusing to readers.

Response: Thank you for the same question as Reviewer 4. We designed three functional modules to *de novo* biosynthesis of 14-hydroxy-dehydroabietic acid (**11**). Module I was used to supply for diterpene miltiradiene (**5**); module II was for dehydroabietic acid (**7**) production and involves subsequent specific discussion of synthetic biology strategies; and the functional module III contained CYP82D274 for production of **11**. We have revised statement of three modules, and hope to get your approval.

Reviewer 2 Comment 13

- Line 500, when interpreting phylogenetic analysis and other evolutionary analyses, it is important to recognize these are inferred possibilities instead of absolute truth.
- Line 523 and 530, genes “were not transcribed” is incorrect. Should say something like “the expression of these genes was too low to be detected”.
- Line 553, The MALDI data should be presented in full in the results section instead of in the discussion.
- I find many figure legends lacking sufficient information necessary for the readers to understand these figures.

Response: Thank you for your careful review of our manuscript, we have made the appropriate changes to the main text and figures based on your suggestions.

- We acknowledge that in interpreting the evolutionary analyses is the possibility of inferring. We have made corrections.
- The MALDI result has been adjusted to the Results section.

Reviewer #3

Reviewer 3 Comment 1

The authors have generated some really interesting data, and shown further evidence that CYPs play a huge role in plant specialized metabolism. The work is significant and very promising and is based on well established models. Overall the paper is fine, though I have some problems that needs to be addressed.

The introduction is very lengthy and provides way too much information. You do not have to justify why to work with triptolide. This is already a well known compound. Thus I suggest to delete the following paragraph (almost 2 pages). Line 25-29; Line 32-44 (then delete however); line 52-56; line 68-75. All this work is redundant.

Response: Thank you for your patient review and guidance of our manuscript, and we have deleted and modified the redundant sentences.

Reviewer 3 Comment 2

The authors should concentrate the introduction on what is discussed. Also I miss some introduction on other CYP's involved in diterpenoid biosynthesis, e.g. forskolin (also made in yeast).

Response: Thank you very much for your suggestion, we have rewritten the Introduction section and added CYPs to the main text.

Reviewer 3 Comment 3

Cytochrome P450 reductases have now for about 10 years been abbreviated POR and NOT CPR like the authors have chosen. Even wikipedia is updated on this. I will require that the authors fix this through out. CPR is used for other enzymes, thus the change.

Response: Thank you for pointing out this problem, it seems that we have not paid much attention to this abbreviation in the past. We have checked in PubMed, using “cytochrome P450 oxidoreductases” as the keyword, and found that most of the POR were for drug or disease CYP (Ref. 1, 2); then used “cytochrome P450 oxidoreductases AND POR AND plant”, there are only a few plant-related papers. Thank you very much for your persistence, which is very important for the promotion of scholarship, and we have revised this paper in its entirety.

Refs:

1. Yan, B. et al. Membrane Damage during Ferroptosis is caused by oxidation of phospholipids catalyzed by the oxidoreductases POR and CYB5R1. *Mol Cell* **81**, 355-369 (2021).
2. Zou Y, Li H, Graham ET, Deik AA, Eaton JK, Wang W, Sandoval-Gomez G, Clish CB, Doench JG, Schreiber SL. Cytochrome P450 oxidoreductase contributes to phospholipid peroxidation in ferroptosis. *Nat Chem Biol* **16**, 302-309 (2020).

Reviewer 3 Comment 4

Specific comments on the results.

Minor thing, but in headings, please use full name of chemical structures e.g. line 380

In line 420-423 it states that CYP82D270 co-expressed with C263 gives higher yield than D270 alone. I cannot find those data - I think this is a crucial point and should be discussed further in connection with the speculative substrate of D263 being 6 and not 7. Here again the authors should consult the work on Forskolin, where similar observations was made.

Line 423 "the cyp728D70 product" - use the number 7 instead of this odd naming.

Response: Thank you for your questions, we have made the appropriate changes. We wrote in the manuscript is “when CYP728B70 was coexpressed with CYP82D263, the yield of **7** was increased compared to CYP728B70 alone, presumably due to the aromatization of **6** by CYP82D263, which provided more substrate for CYP728B70”. The qualitative result was shown in Fig. 6a (the same y-axis results are shown Fig. R2) and the quantitative data were presented in Fig. 5c, indicating that CYP82D263 catalyzed **5** to **6** increased the yield by 25-folds compared to the spontaneous reaction and could provide more substrate for CYP728B70. Hope to get your approval. Thank you!

Fig. R2 GC-MS analysis of products in CYP82D274 and CYP82D263 coexpressed with CYP728B70, respectively.

Reviewer 3 Comment 5

Line 429 - again networks/grid are also seen e.g. in forskolin. This is a link to just one of the papers about this compound. <https://elifesciences.org/articles/23001>

Response: Thank you very much for your comment. We likewise confirmed that biosynthetic pathway of triptolide is not a linear and specific pathway, but rather through metabolic grid, like forskolin and tanshinones.

Reviewer 3 Comment 6

Line 440-465 - PORs are normally divided into two clades POR1 and POR2 (<https://www.sciencedirect.com/science/article/pii/S0031942209004567>) - normally it is observed that the inducible PORs have stronger electron transfer than the constitutive. How does the PORs from *T. wilfordii* fall into these two clades - and please annotate them 1a or 2a respectively of their clade. You can also see more here: <https://www.sciencedirect.com/science/article/pii/S1055790316000294#b0100> Else the POR findings are very interesting and would be nice if this little extra biological information could be added.

Response: Thank you for providing us with literature references. We have analyzed the PORs of *Tripterygium wilfordii* with 54 full-length plant POR sequences for phylogenetic analysis and found that TwPOR1 fell in the CPR2 clade and TwPOR3 and TwPOR4 fell in the CPR1 clade (Fig. R3). In addition, we have added some

biological information of PORs in the manuscript, and more information is detailed in Ref. 2.

Refs:

1. Jensen, K & Møller, BL. Plant NADPH-cytochrome P450 oxidoreductases. *Phytochemistry* 71, 132-141 (2010).
2. Su, P. et al. Probing the Single Key Amino Acid Responsible for the Novel Catalytic Function of ent-Kaurene Oxidase Supported by NADPH-cytochrome P450 reductases in *Tripterygium wilfordii*. *Front Plant Sci* 8, 1756 (2017).

Fig. R3 Phylogenetic analysis of full-length plant CPR sequences.

Line 458, add paragraph change before the picia discussion, please.
As for the strain optimization - consult this thesis for inspiration to the discussion - cannot remember which paper it is in, but the thesis discuss the issue on strain.
https://backend.orbit.dtu.dk/ws/portalfiles/portal/242032750/PhD_thesis_SEBR_TH.pdf

Response: Thank you very much for your providing us with the thesis, which is important to improve the quality of our manuscript and further research.

Reviewer 3 Comment 8

Not sure how the chromosome localization add to this paper. I will suggest to take this part out, and publish by it self. Maybe in Molecular Phylogenetics and Evolution. I really struggle with finding the connection to the rest of the story for this part.

Response: Thank you for your patient review and guidance of our manuscript. We have accepted your suggestions and those of other reviewers, and have made some adjustments to the content of this manuscript.

Our discovery of tandemly duplicated CYP82Ds corroborated their possible important functions in triptolide biosynthesis and we carried out subsequent functional characterization *in planta* and in heterologous yeast. This is one of the important highlights of this paper (Fig. 2), illustrating the genetic story from an evolutionary perspective. We have reduced the redundant descriptions in this section and removed the “mutational analysis” section to narrow the scope of this manuscript, focusing on the discovery of enzymes and establishment of total biosynthetic pathways. In addition, we have complemented the studies *in planta* to verify CYP82D274 and CYP82D263 participate in triptolide biosynthesis through MALDI, metabolite feeding of plant cells, gene overexpression and RNAi (Fig. 4). Hope to get your approval.

1. MALDI.

The 14-hydroxy-dehydroabietic acid (**11**) accumulates in the periderm range (Fig. 4a), which is similar with the distribution pattern of the other acquainted intermediates abietatriene (**6**), dehydroabietic acid (**7**) and triptophenolide (**10**). -- Demonstrate **11** as a potential intermediate from the plant tissue level.

2. Metabolite feeding suspension cells

We have fed **11** to *T. wilfordii* suspension cell cultures and found that the content of triptolide (**1**) and **10** was elevated 35.73 and 35.60 times, respectively, compared with the control (Fig. 4b and Supplementary Data 2). Meanwhile, the catalytic substrate 7 of CYP82D274 and CYP82D263 was sharply reduced. We also detected the expression of biosynthetic pathway genes. -- Direct confirmation that **11** is a precursor of **1**.

3. CYP82D274 and CYP82D263 overexpression in suspension cells

-Since *Tripterygium wilfordii* is a woody vine, we have tried various methods to verify the *in vivo* functions of genes, induction of hairy root and VIGS failed to achieve. We finally used the biotechnology of genegun bombardment¹ and

successfully verified the biochemical functions of CYP728B70² and terpenoid upstream genes^{3,4} *in vivo*.

- Metabolite profiling in *CYP82D274* and *CYP82D263* overexpressing cell lines. The content of **1** and intermediates **10** and **11** increased 1.36 ($P<0.001$), 1.60 ($P<0.05$) and 1.39-fold ($P<0.05$) when *CYP82D274* was overexpressed, and enhanced 1.68 ($P<0.05$), 1.57 ($P<0.05$) and 1.36-fold ($P<0.01$), respectively, under the condition of *CYP82D263* overexpression (Fig. 4d and Supplementary Data 3). --Direct confirmation the involvement of *CYP82D274* and *CYP82D263* in **1** biosynthesis.

-Notably, the content of intermediate 8 decreased to 0.69-fold ($P<0.01$) when *CYP82D274* was overexpressed. --Leads to further exploration of other catalytic processes in *CYP82D274* and *CYP82D263* genes.

4. RNAi

The results showed that the content of final product **1** decreased to 0.38-fold ($P<0.01$) at *CYP82D274*-RNAi and 0.59-fold ($P<0.05$) at *CYP82D263*-RNAi (Fig 4e and Supplementary Data 4). --Direct confirmation the involvement of *CYP82D274* and *CYP82D263* in **1** biosynthesis.

Refs:

1. Zhao, Y. Zhang, Y. Su, P. Yang, J. Huang, L. & Gao, W. Genetic transformation system for woody plant *Tripterygium wilfordii* and its application to product natural celastrol. *Front. Plant Sci.* **8**, 2221 (2017).
2. Tu, L. et al. Genome of *Tripterygium wilfordii* and identification of cytochrome P450 involved in triptolide biosynthesis. *Nat. Commun.* **11**, 971 (2020).
3. Zhang, Y. et al. The expression of *TwDXS* in the MEP pathway specifically affects the accumulation of triptolide. *Physiol. Plant* **169**, 40-48 (2020).
4. Zhang, Y. et al. Overexpression and RNA interference of *TwDXR* regulate the accumulation of terpenoid active ingredients in *Tripterygium wilfordii*. *Biotechnol. Lett.* **40**, 419-425 (2018).

Reviewer 3 Comment 9

Good work, and look forward to see it "in print" ;)

As for the dicussion of 82D as family. Is the conclusion not that 82D hydroxylate aromatic multiple ring compounds and that the sub-familly can be divided into flavonoid clade, coumaring, terpenoid etc - I would love to see the tree (sup frigure 19) as part of the main figures - rather than the chromosome discussion.

Response: Thank you very much for your recognition and review of our manuscript. We have adjusted the writing of this manuscript and moved the phylogenetic analysis to Fig. 7.

Reviewer #4

Reviewer 4 Comment 1

In the manuscript, the authors reported the discovery of two new CYP450 enzymes which were responsible for the 14-hydroxylation of multiple triptolide intermediates. They first analyzed transcriptional data for *T. wilfordii*, triptolide producer, to identify genes co-expressed with other known triptolide pathway genes, which resulted in the discovery of two CYPs that can catalyze the 14-hydroxylation. Then, they tried to analyze these two genes by *in vivo* and *in vitro* experiments with feeding of different precursors as well as mutagenesis studies to explore their mechanism, during which they also found the enzymes can catalyze one step previously known as spontaneous reaction. The authors then tried to perform metabolic engineering to achieve the *de novo* biosynthesis of 14-hydroxy-dehydroabiatic acid in yeast, although the exploration was simple probably due to the page limit. Last, the author tried to study the evolution of these genes through chromosome localization analysis and formed a hypothesis with limited clues. Overall, the discovery of these genes is important, which brings the whole community closer to the complete biosynthesis of triptolide. However, the organization of this manuscript and the description of experiments need a lot of improvements before it can be further considered for publication.

1. The authors tried to study the two enzymes from many different perspectives (mechanistic, structural, evolutionary, biosynthesis, etc.). Although it is good for the audience to know more about the target enzymes, the page/word limit prevent the author from describing all the information in details. It would be good for the authors to narrow the scope of this manuscript, focusing either on the discovery of enzymes and establishment of total biosynthetic pathways, or on the detailed characterization of the evolution/mechanism of these enzymes.

Response: Thank you for your patient review and guidance of our manuscript. We have accepted your and Reviewer 3's suggestion to remove "mutational analysis" section to narrow the scope of this manuscript, focusing on the discovery of enzymes and establishment of total biosynthetic pathways. We have complemented the studies *in planta* to verify CYP82D274 and CYP82D263 participate in triptolide biosynthesis through MALDI, Metabolite feeding, gene overexpression and RNAi (Fig. 4). Hope to get your approval.

1. MALDI.

The 14-hydroxy-dehydroabiatic acid (**11**) accumulates in the periderm range (Fig. 4a), which is similar with the distribution pattern of the other acquainted intermediates abietatriene (**6**), dehydroabiatic acid (**7**) and triptophenolide (**10**). -- Demonstrate **11** as a potential intermediate from the plant tissue level.

2. Metabolite feeding suspension cells

We have fed **11** to *T. wilfordii* suspension cell cultures and found that the content of triptolide (**1**) and **10** was elevated 35.73 and 35.60 times, respectively, compared with the control (Fig. 4b and Supplementary Data 2). Meanwhile, the catalytic substrate **7** of CYP82D274 and CYP82D263 was sharply reduced. We also detected the expression of biosynthetic pathway genes. -- Direct confirmation that **11** is a

precursor of **1**.

3. *CYP82D274* and *CYP82D263* overexpression in suspension cells

-Since *Tripterygium wilfordii* is a woody vine, we have tried various methods to verify the *in vivo* functions of genes, induction of hairy root and VIGS failed to achieve. We finally used the biotechnology of genegun bombardment¹ and successfully verified the biochemical functions of *CYP728B70*² and terpenoid upstream genes^{3,4} *in vivo*.

- Metabolite profiling in *CYP82D274* and *CYP82D263* overexpressing cell lines. The content of **1** and intermediates **10** and **11** increased 1.36 ($P<0.001$), 1.60 ($P<0.05$) and 1.39-fold ($P<0.05$) when *CYP82D274* was overexpressed, and enhanced 1.68 ($P<0.05$), 1.57 ($P<0.05$) and 1.36-fold ($P<0.01$), respectively, under the condition of *CYP82D263* overexpression (Fig. 4d and Supplementary Data 3). --Direct confirmation the involvement of *CYP82D274* and *CYP82D263* in **1** biosynthesis.

-Notably, the content of intermediate 8 decreased to 0.69-fold ($P<0.01$) when *CYP82D274* was overexpressed. --Leads to further exploration of other catalytic processes in *CYP82D274* and *CYP82D263* genes.

4. RNAi

The results showed that the content of final product **1** decreased to 0.38-fold ($P<0.01$) at *CYP82D274*-RNAi and 0.59-fold ($P<0.05$) at *CYP82D263*-RNAi (Fig 4e and Supplementary Data 4). --Direct confirmation the involvement of *CYP82D274* and *CYP82D263* in **1** biosynthesis.

Refs:

1. Zhao, Y. Zhang, Y. Su, P. Yang, J. Huang, L. & Gao, W. Genetic transformation system for woody plant *Tripterygium wilfordii* and its application to product natural celastrol. *Front. Plant Sci.* **8**, 2221 (2017).
2. Tu, L. et al. Genome of *Tripterygium wilfordii* and identification of cytochrome P450 involved in triptolide biosynthesis. *Nat. Commun.* **11**, 971 (2020).
3. Zhang, Y. et al. The expression of *TwDXS* in the MEP pathway specifically affects the accumulation of triptolide. *Physiol. Plant* **169**, 40-48 (2020).
4. Zhang, Y. et al. Overexpression and RNA interference of *TwDXR* regulate the accumulation of terpenoid active ingredients in *Tripterygium wilfordii*. *Biotechnol. Lett.* **40**, 419-425 (2018).

Reviewer 4 Comment 2

2. Similar to point 1, the authors even had original research in their discussion section, which is another sign of lack of space for the overwhelming content.

Response: We have adjusted the MALDI data to the results section (Fig. 4a).

Reviewer 4 Comment 3

3. The authors need improve their writing/figure quality to describe their work clearer, both scientifically and language-wise.

a) Scientific issues:

Line 318: TwCPR3, when first introduce an enzyme, it is important to have a full name for them. Many other genes have the same issue (Line 435,449, etc).

Line 327: which partner CPR?

Line 383: “we transformed plasmids” – which plasmids? Into which engineered yeast?

Line 384: “self-produced” – does it mean self-produced by yeast, or in-house production?

Line 449: How were they truncated?

Line 476: which genes were integrated? Which were on the plasmids?

Response:

- We have added full name of enzymes in the main text. The full name of TwCPR3 is shown in the second section of the Results.

- We have revised to “redox partner TwCPR3”.

- We have added the descriptions of plasmids and engineered yeast.

- “self-produced” mean self-produced by yeast. We have revised the statement.

- CYP728B70 was truncated transit peptides and TwCPRs were truncated transmembrane domain, which were described in detail in the following sentences and Materials and methods section.

- We have added the missing information.

Reviewer 4 Comment 4

b) Language issues:

Line81-82: “via a metabolic network” is difficult to understand. It seems that the authors tried to mean that triptolide biosynthesis is not linear and some specific intermediates can be synthesized from the same precursor through different pathways. If true, the sentence is no clear.

Response: Thank you for pointing out our problems.

- We used “metabolic network” to represent that triptolide biosynthesis is not linear but multi-pathways. It seems “metabolic network” is too big to properly express our views. According to the Ref. 1, we prefer to represent our views in a “metabolic grid” or “multiple pathways”. So, we have changed all the “metabolic network” words in the full text, and hope that we can get your approval.

Refs:

1. Ma, Y. et al. Expansion within the CYP71D subfamily drives the heterocyclization of tanshinones synthesis in *Salvia miltiorrhiza*. *Nat Commun* **12**, 685 (2021).

2. Peters RJ. Uncovering the complex metabolic network underlying diterpenoid phytoalexin biosynthesis in rice and other cereal crop plants. *Phytochemistry* **67**, 2307-2317 (2006).

3. Tohge, T. et al. Exploiting natural variation in tomato to define pathway structure

and metabolic regulation of fruit polyphenolics in the lycopersicum complex. *Mol Plant* **13**, 1027-1046 (2020).

Reviewer 4 Comment 5

Line86: “located on the length of the 133 kb” is not understandable.
Line297: “share” the biosynthesis through coexpression is not clear.

Response:

- Thank you very much for the same question as Reviewer 2, we have revised it to “we found that two CYP82Ds located in a 133 kb tandem duplication cluster of CYP82Ds”.
- We have revised the sentence to “Pathway genes undertake the biosynthesis of secondary metabolites together, and these genes often exhibit similar expression patterns”.

Reviewer 4 Comment 6

Line 415: ”three generation modules” is not clear.

Response: Thank you for the same question as Reviewer 2. We designed three functional modules or metabolic modules to *de novo* biosynthesis of 14-hydroxy-dehydroabiatic acid (**11**). Module I was used to supply for diterpene miltiradiene (**5**); module II was for dehydroabiatic acid (**7**) production and involves subsequent specific discussion of synthetic biology strategies; and the functional module III contained CYP82D274 for production of **11**. We have revised statement of three modules, and hope to get your approval.

Reviewer 4 Comment 7

Line339-350: This paragraph is in isolation, the authors need to integrate it with the next paragraph.
Line366: “even though the A326S ... products disappeared” is not a clear sentence.

Response: Thank you for your suggestions. We have removed the “mutational analysis” section.

Reviewer 4 Comment 8

c) Figure issues:
Fig1c: difficult to draw the conclusion of “same pattern” from the figure.

Response: Thank you for pointing out the errors in the figures. We analyzed 416

CYPs in the *T. wilfordii* transcriptome for expression correlation with previously characterized genes involved in the biosynthetic pathway of triptolide. The Fig. 1c represents the most relevant genes which were clustered together with functional genes, and that's why we use "same pattern...". For complete 67 clusters please see Source Data, which we have also described in the manuscript.

Reviewer 4 Comment 9

Fig2b: It should be proposed catalytic reaction.

Response: The product **11** was enriched and identified as 14-hydroxy-dehydroabiatic acid (Supplementary Figs. S3-S7 and Supplementary Note). So, the Fig. 3b (new figure number) is a solid catalytic reaction.

Reviewer 4 Comment 10

FigS10, Fig3f and more: it would be better to label every sub-spectra.

Response: Thank you for your suggestion, and we have labeled every sub-spectra.

Reviewer 4 Comment 11

Fig4a: whether all spectrum have the same y-axis? It seems not.

Response: Thank you for pointing out your question, each Y-axis is indeed different. We drew the Fig. 6a with the aim of investigating the ability of yeast to simultaneously express multiple CYPs and generated product **11**. The results for the same y-axis are shown below (Fig. R2), and for the sake of neatness of the figure, Fig. 6a prefers to be a qualitative result, while quantitative data are presented in Fig. 6b and 6c. Hope to get your approval. Thank you!

Fig. R2 GC-MS analysis of products in CYP82D274 and CYP82D263 coexpressed with CYP728B70, respectively.

Reviewer 4 Comment 12

Minor points:

1. Better to have atom number of one of the molecules to help the audience understand all the positions.
2. Line 459: why not select CYP720B4 first?

Response: 1. We originally labeled each atom number in Fig. 6, but have now advanced the numbering to Fig. 5d.

2. We first used CYP728B70 of the same species as CYP82Ds and then replaced CYP720B4 of the other species, and also compared the difference in the yield of **11** between the two enzymes, using a synthetic biology strategy with progressively higher yields.

Reviewers' Comments:

Reviewer #1:

Remarks to the Author:

My comments are addressed.

Reviewer #2:

Remarks to the Author:

In this revised version of the manuscript, the authors carried out a number of additional experiments trying to address the criticisms that I raised in the last round of review. However, many of these experiments were not designed properly (lacking key controls), and the results were often misinterpreted. As a result, the manuscript was not much improved.

1. Feeding of 11 to *T. wilfordii* suspension cell cultures led to elevated accumulation of 1 and 10 does not conform that 11 is a precursor of 1. Adding 11 to *T. wilfordii* suspension cell cultures may simply elicit de novo biosynthesis of 1. The fact that 7 is greatly reduced upon addition of 11 suggests changing physiology of the cell culture. The experiment could be greatly improved by using isotopically labeled 11, and show tracing of the isotope in the final product.

2. The overexpression experiment is problematic. Fig. 4c shows PCR amplification of a fragment of the vector. The expression levels of the overexpressed P450 genes were not shown. Again, overexpression of CYP82D274 and CYP82D263 leading to elevated 1 biosynthesis is not a direct confirmation that these two genes are directly involved in the biosynthesis of 1.

3. The RNAi experiment is also problematic. Close examination of Fig. S10 indicates that CYP82D274 RNAi did not work in repressing CYP82D274, whereas CYP82D263 RNAi construct also repressed CYP82D274 expression. Statement like "In addition, excluding differences in biological replicates, we could still conclude a trend towards reduced biosynthesis of intermediates like 10, 8, 11 except for the accumulation of substrate 7." Is simply not acceptable. The fact that 11 is less affected than 1 in RNAi lines may suggest CYP82D263 and CYP82D274's in vivo catalytic functions may be different than what the authors showed in vitro (assuming the RNAi experiments worked as intended).

4. There are many grammar issues in writing throughout the manuscript which I am not detailing here.

Reviewer #3:

Remarks to the Author:

Thanks for updating the manuscript. I find that this is much better now. Overall the paper is much much better now.

Reviewer #4:

Remarks to the Author:

The authors have addressed most of the constructive issues by me and other reviewers. They have added experiments such as substrate feeding in suspension cells as well as RNAi to further confirm the gene function in plant. The manuscript is now in a better shape and ready for publication after fixing the following minor issues:

1. "self-produce" is still miss-leading, if the strain can produce 5, then the author should introduce the strain when it first appears and then they can use the strain name instead of "strain self-produce 5".

2. For the transit peptide truncation, why 50 amino acids? Is it predicted by a software or it's an arbitrary number? There are many tools can predict transit peptide (or signal peptide as a better annotation), such as SignalP

3. I still have concern with fig1c, I understand the authors have screened many genes to find these few genes have "similar expression patterns" with other pathway genes. But I suggest the authors to explain it more clearly. For example, 1165.2, 1165.1, 1159.1 and more genes have significant higher Z-score in SB2 comparing with all the TPSs. How indeed did the author calculate the similarity?

Response to reviewers

Dear Reviews,

We are very grateful for your valuable comments and suggestions which improve the quality of this manuscript. We have carefully taken your comments into consideration in preparing our revision and have made modifications and corrections. Here below is our description of revision, and we sincerely hope it meets with your approval.

Reviewer #1

Reviewer 1 Comment 1

My comments are addressed.

Response: Thank you for your review and approval. We are honored to have your guidance.

Reviewer #2

Reviewer 2 Comment 1

In this revised version of the manuscript, the authors carried out a number of additional experiments trying to address the criticisms that I raised in the last round of review. However, many of these experiments were not designed properly (lacking key controls), and the results were often misinterpreted. As a result, the manuscript was not much improved.

1. Feeding of 11 to *T. wilfordii* suspension cell cultures led to elevated accumulation of 1 and 10 does not conform that 11 is a precursor of 1. Adding 11 to *T. wilfordii* suspension cell cultures may simply elicit de novo biosynthesis of 1. The fact that 7 is greatly reduced upon addition of 11 suggests changing physiology of the cell culture. The experiment could be greatly improved by using isotopically labeled 11, and show tracing of the isotope in the final product.

Response: Thank you for your patient review and guidance of our manuscript. We have adjusted the content of this manuscript based on your suggestions and hope to get your approval.

Firstly, we have determined the changes in triptolide (**1**) and its intermediates after RNAi, and the results indicated that *CYP82D274* and *CYP82D263* are involved in the biosynthesis of **1**. Subsequently, changes in metabolites and genes were verified using MALDI, feeding and gene overexpression, which then led to further characterization of the catalytic function of the two CYPs on substrates **5**, **6**, **8** in the following sections.

Going back to this question, as you said, the best way is to use isotopically labeled **11**, and we have been trying this approach, but because *Tripterygium*

wilfordii is a woody vine, its genetic transformation system is very difficult to establish and needs to be fed with large amounts of metabolite, and we are limited by low yields and high costs, both biosynthetically and chemically. So we turned to experiments using unlabeled metabolite **11** in the context of having identified it as an intermediate. This approach can also be found in the following references.

Regarding the dramatic decrease in substrate **7** after feeding **11**. We have detected that genes known to be involved in **1** biosynthesis, *CYP82D274*, *CYP82D263*, *CYP728B70*, *TPS7(v2)*, *TPS27(v2)* and *TwGGPPS*, exhibited varying proportions of 0.42- to 0.76-fold decrease in expression, but not in genes of the MVA and MEP pathways (Fig. 4b). This result can be well explained by the fact that the increase of **11** decreased the expression of each enzyme in its previous steps and reduced the accumulation of **7**, but provided sufficient substrate for biosynthesis to advance toward **1**.

Refs:

1. Su, P. et al. Identification and functional characterization of diterpene synthases for triptolide biosynthesis from *Tripterygium wilfordii*. *Plant Journal*. **93**, 50-65 (2018).
2. Zhou, J. et al. Friedelane-type triterpene cyclase in celastrol biosynthesis from *Tripterygium wilfordii* and its application for triterpenes biosynthesis in yeast. *New phytologist*. **223**, 722-735 (2019).
3. Hofer, R. et al. Geraniol hydroxylase and hydroxygeraniol oxidase activities of the CYP76 family of cytochrome P450 enzymes and potential for engineering the early steps of the (seco)iridoid pathway. *Metabolic Engineering*. **20**, 221-232 (2013).
4. Ohnishi T, et al. C-23 hydroxylation by Arabidopsis CYP90C1 and CYP90D1 reveals a novel shortcut in brassinosteroid biosynthesis. *Plant Cell*. **18**, 3275-3288 (2006).
5. Choe, S. et al. The DWF4 gene of Arabidopsis encodes a cytochrome P450 that mediates multiple 22alpha-hydroxylation steps in brassinosteroid biosynthesis. *Plant Cell*. **10**, 231-243 (1998).

Reviewer 2 Comment 2

2. The overexpression experiment is problematic. Fig. 4c shows PCR amplification of a fragment of the vector. The expression levels of the overexpressed P450 genes were not shown. Again, overexpression of *CYP82D274* and *CYP82D263* leading to elevated **1** biosynthesis is not a direct confirmation that these two genes are directly involved in the biosynthesis of **1**.

Response: We have supplemented the gene overexpression data in Supplementary Fig. S10b. *CYP82D274* and *CYP82D263* were increased to 2.70- and 3.14-fold, respectively. By studying the changes in each metabolite after gene overexpression, we have observed a significant decrease in the amount of intermediate **8** and an increase in the amount of **15** upon overexpression of *CYP82D274*, which in turn further characterized the catalytic functions of the two CYPs for substrates **5**, **6**, and **8** in the following.

Reviewer 2 Comment 3

3. The RNAi experiment is also problematic. Close examination of Fig. S10 indicates that CYP82D274 RNAi did not work in repressing CYP82D274, whereas CYP82D263 RNAi construct also repressed CYP82D274 expression. Statement like “In addition, excluding differences in biological replicates, we could still conclude a trend towards reduced biosynthesis of intermediates like 10, 8, 11 except for the accumulation of substrate 7.” Is simply not acceptable. The fact that 11 is less affected than 1 in RNAi lines may suggest CYP82D263 and CYP82D274’s *in vivo* catalytic functions may be different than what the authors showed *in vitro* (assuming the RNAi experiments worked as intended).

Response: We really appreciate you for pointing out this problem.

Cells of woody plant *T. wilfordii* origin are larger than other plant suspension cells, and each cell is relatively independent, resulting in larger errors between biological replicates. The results of *CYP82D274* and *CYP82D263* RNAi studies, which have been optimized during the past two months, are shown in Fig. 3c.

The expression of *CYP82D274* or *CYP82D263* was significantly decreased only in the respective sample groups, indicating that they were targeted for disruption as expected.

We have indicated significant changes in the levels of four metabolites and expression of known pathway genes under conditions of CYP RNAi and suggested that *CYP82D274* and *CYP82D263* are involved in the biosynthesis of **1**. In addition, we have experimentally confirmed that the metabolic grid of CYP82D274/263 is a multi-substrate catalytic process, and **11** is only one of the products, so the inhibition of **1** after gene RNAi is more pronounced than **11**, consistent with the catalytic process we have shown *in vitro*.

Reviewer 2 Comment 4

4. There are many grammar issues in writing throughout the manuscript which I am not detailing here.

Response: We regret there were problems with the English. The paper has been carefully revised by a professional language editing service (order No. WQXGCQ5C) to improve the grammar and readability.

Reviewer #3

Reviewer 3 Comment 1

Thanks for updating the manuscript. I find that this is much better now. Overall the paper is much much better now.

Response: Thanks a lot for your kind comments and we are honored to have your guidance.

Reviewer #4

Reviewer 4 Comment 1

The authors have addressed most of the constructive issues by me and other reviewers. They have added experiments such as substrate feeding in suspension cells as well as RNAi to further confirm the gene function in plant. The manuscript is now in a better shape and ready for publication after fixing the following minor issues:

1. “self-produce” is still miss-leading, if the strain can produce 5, then the author should introduce the strain when it first appears and then they can use the strain name instead of “strain self-produce 5”.

Response: Thank you for your patient review and kind comments of our manuscript. We have removed the misleading “self-produce” and revised the statement.

Reviewer 4 Comment 2

2. For the transit peptide truncation, why 50 amino acids? Is it predicted by a software or it's an arbitrary number? There are many tools can predict transit peptide (or signal peptide as a better annotation), such as SignalP

Response: We used the ChloroP 1.1 Server (<http://www.cbs.dtu.dk/services/ChloroP/>) to predict that the chloroplast transit peptide (cTP) for CYP728B70 was 50 amino acids.

We have added predictive tools in the main text.

Reviewer 4 Comment 3

3. I still have concern with fig1c, I understand the authors have screened many genes to find these few genes have “similar expression patterns” with other pathway genes. But I suggest the authors to explain it more clearly. For example, 1165.2, 1165.1, 1159.1 and more genes have significant higher Z-score in SB2 comparing with all the TPSs. How indeed did the author calculate the similarity?

Response: Thank you for your comments, which have enabled us to greatly improve the quality of this manuscript. In fact, we have discovered “38 CYP genes from 16 CYP families exhibited the same pattern of root periderm-specific high expression and clustered with identified functional genes (Fig. 1c)”. Significantly, the most numerous (about 13%) genes were from the CYP82D subfamily, compared to other subfamilies, this CYP82D subfamily which had to attract our attention. In addition, we have modified Fig. 1c to make the whole story of CYP82D274 clearer.

Overall, thanks a lot for your insightful comments and we are honored to have your guidance.

Reviewers' Comments:

Reviewer #2:

Remarks to the Author:

Although I recognize that some experiments involving *T. wilfordii* plants or derived cells are challenging, comprised experimental designs and less-than-ideal results are difficult to interpret. Therefore, confusions and explanations based on these problematic experiments are simply speculative. The exact role of CYP82D274 and CYP82D263 in triptolide biosynthesis remain elusive.

Response to Reviewer 2

Comment 1

Although I recognize that some experiments involving *T. wilfordii* plants or derived cells are challenging, comprised experimental designs and less-than-ideal results are difficult to interpret. Therefore, confusions and explanations based on these problematic experiments are simply speculative. The exact role of CYP82D274 and CYP82D263 in triptolide biosynthesis remain elusive.

Response:

We would like to express our sincere respect to you. Your rigorous pursuit of scientific research makes us constantly improve the logic and content of this manuscript. We have supplemented the kinetic analysis and isotope labeling experiments under your valuable suggestions. Below is our description of the revisions, and we sincerely hope it meets with your approval.

We selected ^{18}O as the isotope marker and conducted *in vitro* enzymatic reactions according to the kinetic profiles to obtain ^{18}O -labeled 14-hydroxy-dehydroabietic acid (**11**). The purified ^{18}O -**11** was fed into cells that interfered with TPS7&27 and blocked the MEP pathway, and MeJA was subsequently used to induce gene expression to increase downstream metabolic flow. Equal concentrations of unlabeled **11** with identical treatment and WT cells were used as controls. Each group has three biological replicates. Although the native metabolic pathway was not completely inhibited, we still found the peak of ^{18}O -triptinin B (**9**) (Fig. 1 below), which could be further lactonized by CYP71BE to generate acquainted intermediate triptophenolide (**10**)¹. The isotope labeling result indicated that **11** is a precursor of **1**.

In another recently published article¹, it is illustrated that the minimal composition of four CYPs generates the analog triptonide of **1**, in which key enzymes for downstream lactone ring and triepoxide formation are detailed.

Fig. 1 Metabolite analysis of stable-isotope labeling from duplicate cell samples. Wild type (WT) and unlabeled-**11** treatments were used as controls. ^{18}O -**11** was converted into ^{18}O -triptinin B (**9**) in plant cell, and mass spectrum of labeled versus unlabeled metabolites were provided.

In this version, we have adjusted the logic and content of this manuscript, and the section on functional characterization is divided into the following points:

1. C-14 hydroxylation

CYP82D274 and CYP82D263 catalyzed the C-14 hydroxylation of dehydroabietic acid (**7**) in *Saccharomyces cerevisiae* (Fig. 3a).

2. In vivo assays

Targeted RNAi assays in plant cells that inhibition of *CYP82D274* and *CYP82D263* transcriptional expression resulted in significant reductions in triptolide (**1**) accumulation by 60% and 47%, respectively, as well as significant reductions in direct product **11**, intermediates triptobenzene D (**8**) and triptophenolide (**10**) (Fig. 4d). When *CYP82D274* and *CYP82D263* were overexpressed (Supplementary Fig. 11b), a significant increase in the accumulation of metabolites **1**, **10**, and **11** was observed (Fig. 4e). The results of these *in vivo* assays suggested that CYP82D274 and CYP82D263 are involved in **1** biosynthesis.

3. 14-hydroxy-dehydroabietic acid is a precursor of triptolide

We used isotopically labeled or unlabeled metabolites to feed *Tripterygium wilfordii* suspension cells. When feeding unlabeled-**11** to cell cultures and found that the content of **1** and **10** was elevated 35.73 and 35.60 times, respectively, compared with the control (Fig. 4b). The expression of all downstream pathway genes known to be involved in **1** biosynthesis (*CYP82D274*, *CYP82D263*, *CYP728B70*, *TPS7(v2)*, *TPS27(v2)*, and *TwGGPPS*) reduced to 0.42-0.76 folds, but did not affect genes of the upstream MVA and MEP pathways (Fig. 4b). Subsequently, the purified ¹⁸O-**11** fed samples arose a peak of ¹⁸O-labeled triptinin B (**9**) (Fig. 4c), which could be further lactonized and triepoxidation by CYP71BE and CYP82D to generate **10** and **1**¹.

4. C-14 hydroxylation grid and aromatization

Changes in the concentration of several metabolites were revealed by CYP82Ds overexpression (Fig. 4e), thus revealing that CYP82D274 catalyzes C-14 hydroxylation of intermediates **6**, **7**, and **8**. Both CYP82D27 and CYP82D263 catalyzed the aromatization of miltiradiene (**5**), which has been repeatedly reported as a spontaneous process (Fig. 5).

5. Kinetic analysis indicates CYP82D274 is more affinity to (7)

The K_m of CYP82D274 and CYP82D263 for catalyzing **7** to be $0.99 \pm 0.17 \mu\text{M}$ and $8.42 \pm 1.89 \mu\text{M}$, respectively. The K_m of CYP82D274 catalyzing **5** was $47.30 \pm 12.98 \mu\text{M}$. No product was detected when CYP82D274 catalyzed **8** *in vitro*. Deep learning-based DLKcat² was employed for K_{cat} prediction, and the results showed that the turnover numbers of CYP82D274 were 4.9260 s^{-1} (for substrate **7**), 4.6322 s^{-1} (for **6**), 3.5838 s^{-1} (for **8**) and 1.5567 s^{-1} (for **5**), respectively.

Reference

1. Hansen, N.L. et al. Tripterygium wilfordii cytochrome P450s catalyze the methyl shift and epoxidations in the biosynthesis of triptonide. *Nature Communications* **13** (2022).
2. Li, F. et al. Deep learning-based kcat prediction enables improved enzyme-constrained model reconstruction. *Nature Catalysis* **5**, 662-672 (2022).

\Reviewers' Comments:

Reviewer #2:

Remarks to the Author:

Many aspects of the paper have been improved at this point, but the manuscript is still littered with many awkward sentences, descriptions and unclear presentations (also in a few figures and figure legends). Proofreading by other colleagues and/or retaining professional editing service are recommended.

I can only list a few examples below as I don't have the capacity to exhaustively list them:

1. Line 12, "indicate a significant affinity"
2. Line 13, "The precursor 14-hydroxy-dehydroabiatic acid has succeeded in producing and improving its yield with *Saccharomyces cerevisiae*."
3. Line 34, "only 0.0001%". On what basis?
4. Line 150, "was extremely superior to".
5. Line 194, "catalysate"
6. Line 243, "and is more affinity to"
7. Figure 4c, as currently presented, is uninterpretable. Please reconsider how substrates and products (labeled vs unlabeled) are indicated on the traces. The figure legends also need to be more self-explanatory.
8. I found the bar graphs for gene expression hard to digest. More description in the figure legends would help.
9. The use of DLKcat for kcat prediction is highly controversial. Without validation, the predicted values are irrelevant, and add little value to the story. I suggest the authors report the Vmax deduced from the kinetic assays.

Response to Reviewer 2

Dear Reviewer,

We would like to express our sincere respect to you. Your rigorous pursuit of scientific research makes us constantly improve the content of this manuscript. Here below is our description of revision, and we sincerely hope it meets with your approval. The Chinese Year of the Rabbit is coming. We wish you a Happy Chinese New Year.

Comment 1

Many aspects of the paper have been improved at this point, but the manuscript is still littered with many awkward sentences, descriptions and unclear presentations (also in a few figures and figure legends). Proofreading by other colleagues and/or retaining professional editing service are recommended.

I can only list a few examples below as I don't have the capacity to exhaustively list them:

1. Line 12, "indicate a significant affinity"
2. Line 13, "The precursor 14-hydroxy-dehydroabiatic acid has succeeded in producing and improving its yield with *Saccharomyces cerevisiae*."
4. Line 150, "was extremely superior to".
5. Line 194, "catalysate"
6. Line 243, "and is more affinity to"

Response: We appreciate the time you took to read our revised manuscript and provide critical comments. We have revised inappropriate language throughout the article.

Comment 2

3. Line 34, "only 0.0001%". On what basis?

Response: We have modified to the actual content of triptolide in the root.

Comment 3

7. Figure 4c, as currently presented, is uninterpretable. Please reconsider how substrates and products (labeled vs unlabeled) are indicated on the traces. The figure legends also need to be more self-explanatory.

8. I found the bar graphs for gene expression hard to digest. More description in the figure legends would help.

Response: Thanks for your suggestions, we have redrawn Figure 4c and added the description of figure legends.

Comment 4

9. The use of DLKcat for kcat prediction is highly controversial. Without validation, the predicted values are irrelevant, and add little value to the story. I suggest the authors report the V_{max} deduced from the kinetic assays.

Response: Thank you for your kind advice, we have increased the data of V_{max} and removed the predicted K_{cat} .